# STABILITY AND CONVERGENCE THEORY FOR LEARNING RESNET: A FULL CHARACTERIZATION

## ABSTRACT

ResNet structure has achieved great success since its debut. In this paper, we study the stability of learning ResNet. Specifically, we consider the ResNet block $h_l = \phi(h_{l-1} + \tau \cdot g(h_{l-1}))$ where $\phi(\cdot)$ is ReLU activation and $\tau$ is a scalar. We show that for standard initialization used in practice, $\tau = 1/\Omega(\sqrt{L})$ is a sharp value in characterizing the stability of forward/backward process of ResNet, where $L$ is the number of residual blocks. Specifically, stability is guaranteed for $\tau \leq 1/\Omega(\sqrt{L})$ while conversely forward process explodes when $\tau > L^{-\frac{1}{2}+c}$ for a positive constant $c$. Moreover, if ResNet is properly over-parameterized, we show for $\tau \leq 1/\tilde{\Omega}(\sqrt{L})$ gradient descent is guaranteed to find the global minima [1], which significantly enlarges the range of $\tau \leq 1/\tilde{\Omega}(L)$ that admits global convergence in previous work. We also demonstrate that the over-parameterization requirement of ResNet only weakly depends on the depth, which corroborates the advantage of ResNet over vanilla feedforward network. Empirically, with $\tau \leq 1/\sqrt{L}$, deep ResNet can be easily trained even without normalization layer. Moreover, adding $\tau = 1/\sqrt{L}$ can also improve the performance of ResNet with normalization layer.

## 1 INTRODUCTION

Residual Network (ResNet) has achieved great success in computer vision tasks since the seminal paper (He et al., 2016). Moreover, the ResNet structure has also been extended to natural language processing and achieved the state-of-the-art performance (Vaswani et al., 2017; Devlin et al., 2018). In this paper, we study the forward/backward stability and convergence theory of learning ResNet.

Specifically, we consider the ResNet with the following residual block,

$$h_l = \phi(h_{l-1} + \tau \boldsymbol{W}_l h_{l-1}), \qquad (1)$$

where $\phi(\cdot)$ is the ReLU activation, $h_l$ is the output of layer $l$, $\boldsymbol{W}_l$ is the parameter of layer $l$ and $\tau$ is a scale factor on the parametric branch in a residual block. We note that standard initialization schemes, e.g., the Kaiming's initialization, are designed to keep the forward and backward variance constant when passing through one layer. However, things become different for ResNet. If $\boldsymbol{W}_l$ adopts Kaiming's initialization, a small $\tau$ is necessary for a stable forward process of ResNet, because the output explodes in expectation for $\tau = 1$ when $L$ gets large. On the other side, a limit form of *Euler's constant* indicates that $\tau = 1/\Omega(L)$ is sufficient for the forward stability as shown in previous work (Allen-Zhu et al., 2019; Du et al., 2019). It is natural to ask

"Are there other values of $\tau$ that can guarantee the stability of ResNet with arbitrary depth?"

We target the above question and unveil that $\tau = 1/\sqrt{L}$ is a sharp value in terms of characterizing the stability of forward/backward process of ResNet with a non-asymptotic analysis. Specifically, stability is guaranteed for $\tau \leq 1/\Omega(\sqrt{L})$. Conversely, for $\tau > L^{-\frac{1}{2}+c}$ the network output grows at least with rate $L^c$ in expectation, which implies forward/backward explosion for large $L$.

One step further, based on the stability argument, we show that if the network is properly over-parameterized, gradient descent is guaranteed to find global minima for training ResNet with $\tau \leq 1/\tilde{\Omega}(\sqrt{L})$, where the range of $\tau$ is significantly enlarged compared to the result in Allen-Zhu et al.

---

[1]We use $\tilde{\Omega}(\cdot)$ to hide logarithmic factor.

(2018b); Du et al. (2018) with $\tau \leq 1/\Omega(L)$. Over-parameterization has been recently used as a hammer to tackle the optimization property (Allen-Zhu et al., 2018b; Du et al., 2018; Zou et al., 2018; Zou and Gu, 2019) of neural network. It considers the case when the neural network is very wide at each layer, which enables the theoretical analysis via the statistical concentration property of the parameter matrix. We show that the over-parameterization requirement of ResNet only weakly depends on the depth, which justifies the advantage of ResNet over feedforward network. Our contribution can be summarized as follows.

- We establish a non-asymptotic analysis showing that $\tau = 1/\sqrt{L}$ is tight in the order sense for characterizing the stability of ResNet.
- For $\tau \leq 1/\Omega(\sqrt{L}\log m)$, we establish the convergence of gradient descent to global minima for learning over-parameterized ResNet with arbitrary depth.

The key step to prove our first claim is a new bound of the spectral norm of the forward process for ResNet with $\tau \leq 1/\Omega(\sqrt{L})$. This bound is a bit surprising as a natural bound $(1+1/\sqrt{L})^L$ explodes. We use martingale theory to characterize the largest possible change after multiple residual mappings, which is shown to be bounded given $\tau < 1/\sqrt{L}$. This technique may be of independent interest for other problems.

The idea for proving global convergence is as follows. First we establish the forward/backward stability at the initialization and derive the gradient upper/lower bounds by utilizing the statistical concentration of random matrix. Then we show the gradient bounds do not change much after perturbation as long as the perturbation is relatively small. Finally by properly choosing the step size, gradient descent update indeed produces small enough perturbation. Besides the forward/backward stability, the key step to show our global convergence is a new gradient upper bound which is tighter than previous works by exploiting the scaling factor $\tau$. This new upper bound enables the weak depth-dependent argument of learning ResNet.

We note that although the global convergence of gradient descent for learning ResNet has been established by Allen-Zhu et al. (2018b); Du et al. (2018) for $\tau = 1/\tilde{\Omega}(L)$, if we train practical ResNet with $\tau = 1/L$, the performance is largely worse than ResNet with normalization layer that is used in practice. In contrast, our stability and convergence theory can serve as a guide for practice. Empirically, we demonstrate that with $\tau = 1/\sqrt{L}$, ResNet can be effectively trained without the normalization layers. Moreover, even with normalization layer, deep ResNet does not perform well in practice. We illustrate the reason from the stability perspective and demonstrate that adding $\tau = 1/\sqrt{L}$ on top of the normalization layer can obtain considerable performance improvement.

## 1.1 RELATED WORKS

A concurrent work Arpit et al. (2019) considers a similar residual mapping and suggests that $\tau = 1/\sqrt{L}$ can keep the norm of the forward/backward pass roughly constant in expectation. In sharp contrast, we consider the standard initialization scheme used in practice and provide rigorous non-asymptotic analysis for the stability for the forward/backward pass with $\tau = 1/\Omega(\sqrt{L})$. Zhang et al. (2019a) demonstrates that ResNet can be trained without normalization layer if the first mapping in a residual block is initialized down by $1/\sqrt{L}$ and the last mapping is initialized to 0. In contrast, our result shows that $\tau = 1/\sqrt{L}$ is sufficient for training a standard ResNet without normalization layer and achieve better empirical performance than Zhang et al. (2019a). Zhang et al. (2019b) argues that a small $\tau$ can give robust ResNet from a partial differential equation interpretation of ResNet but does not provide how small $\tau$ should be. Moreover, our paper is also related to the work that have studied the benefit of ResNet (Veit et al., 2016; Zhang et al., 2018; Hardt and Ma, 2016; Orhan and Pitkow, 2018).

Our paper is closely related to recent work on over-parameterization/neural tanget kernel (NTK) technique. The NTK approach considers the case when the neural network is very wide (often more neurons than samples) at each layer and the training iterates fall into a small region near the initialization. Jacot et al. (2018); Allen-Zhu et al. (2018b); Du et al. (2018); Chizat and Bach (2018); Zou et al. (2018); Zou and Gu (2019); Arora et al. (2019a); Oymak and Soltanolkotabi (2019) showed that gradient descent converges linearly to the global minima for training over-parameterized deep neural network from the optimization perspective. Brutzkus et al. (2017); Li and Liang (2018);

Allen-Zhu et al. (2018a); Arora et al. (2019b); Cao and Gu (2019); Neyshabur et al. (2019) establish the generalization properties of over-parameterized neural network. On the other side, Ghorbani et al. (2019); Chizat et al. (2019); Yehudai and Shamir (2019); Allen-Zhu and Li (2019) discuss the limitation of the NTK approach in characterizing the behavior of neural network. The "limitation" lies in that kernel approach cannot approximate a single ReLU neuron efficiently and the empirical generalization gap between the NTK approach and neural network, which does not underrate our contribution as we focus on the convergence behavior of gradient descent. Moreover, our stability result on the range of $\tau$ holds beyond the NTK regime.

## 1.2 PAPER ORGANIZATION

The rest of this paper is organized as follows. Section 2 introduces the model and notations. Section 3 gives a non-asymptotic analysis on the forward/backward stability of ResNet for a full range of $\tau$. Section 4 presents the global convergence of gradient descent for training over-parameterized ResNet, including the proof roadmap. Section 5 gives some experiments that support our theory. Finally, we conclude in Section 6.

## 2 PRELIMINARIES

There are many residual network models since the seminal paper (He et al., 2016). Here we study a simple ResNet model with *Kaiming's initialization* (He et al., 2016) is described as follows[2],

- Input layer: $h_0 = \phi(\boldsymbol{A}x)$;
- $L - 1$ residual layers: $h_l = \phi(h_{l-1} + \tau \boldsymbol{W}_l h_{l-1})$;
- A fully-connected layer: $h_L = \phi(\boldsymbol{W}_L h_{L-1})$;
- Output layer: $y = \boldsymbol{B} h_L$;
- Initialization: $\boldsymbol{A} \in \mathbb{R}^{m \times p}, \boldsymbol{B} \in \mathbb{R}^{d \times m}$ and $\boldsymbol{W}_l \in \mathbb{R}^{m \times m}$ for $l = [L]$ where entries of $\boldsymbol{A}, \boldsymbol{B}$ and $\boldsymbol{W}_l$ are independently sampled from $\mathcal{N}(0, \frac{2}{m}), \mathcal{N}(0, \frac{2}{d})$ and $\mathcal{N}(0, \frac{2}{m})$, respectively;

where $\phi(\cdot)$ is the ReLU activation function $\phi(\cdot) := \max\{0, \cdot\}$. Specifically, we assume the input dimension is $p$ and hence $x \in \mathbb{R}^p$, the intermediate layers have the same width $m$, and hence $h_l \in \mathbb{R}^m$ for $l = 0, 1, ..., L$, and the output has dimension $d$ and hence $y \in \mathbb{R}^d$. Denote the values before activation by $g_0 = \boldsymbol{A}x, g_l = h_{l-1} + \tau \boldsymbol{W}_l h_{l-1}$ for $l = 1, 2, ..., L-1$ and $g_L = \boldsymbol{W}_L h_{L-1}$. Use $h_{i,l}$ and $g_{i,l}$ to denote the value of $h_l$ and $g_l$, respectively, when the input vector is $x_i$, and $\boldsymbol{D}_{i,l}$ the diagonal sign matrix where $[\boldsymbol{D}_{i,l}]_{k,k} = \mathbf{1}_{\{(g_{i,l})_k \geq 0\}}$.

We introduce a notation $\overrightarrow{\boldsymbol{W}} := (\boldsymbol{W}_1, \boldsymbol{W}_2, \ldots, \boldsymbol{W}_L)$ to represent all the trainable parameters. Throughout the paper, we use $\|v\|$ to denote the $l_2$ norm of the vector $v$. We further use $\|\boldsymbol{M}\|_2$ and $\|\boldsymbol{M}\|_F$ to denote the spectral norm and the Frobenius norm of the matrix $\boldsymbol{M}$, respectively. Denote $\|\overrightarrow{\boldsymbol{W}}\|_2 := \max_{l \in [L]} \|\boldsymbol{W}_l\|_2$ and $\|\boldsymbol{W}_{[L-1]}\|_2 := \max_{l \in [L-1]} \|\boldsymbol{W}_l\|_2$.

The training data set is $\{(x_i, y_i^*)\}_{i=1}^n$, where $x_i$ is the feature vector and $y_i^*$ is the target signal for all $i = 1, ..., n$. We consider the objective function is

$$F(\overrightarrow{\boldsymbol{W}}) := \sum_{i=1}^n F_i(\overrightarrow{\boldsymbol{W}}), \quad \text{where} \ \ F_i(\overrightarrow{\boldsymbol{W}}) := \ell(\boldsymbol{B} h_{i,L}, y_i^*),$$

where $\ell(\cdot)$ is the loss function that measures the discrepancy between the the network output and the target signal. The model is trained by running the gradient descent algorithm. Though ReLU is nonsmooth, we abuse the word "gradient" to represent the value computed through back-propagation.

## 3 FORWARD AND BACKWARD STABILITY OF RESNET

In this section, we establish the stability of training ResNet. We show that when $\tau \leq 1/\Omega(\sqrt{L})$ the forward and backward pass is bounded at the initialization and after small perturbation. On the

---

[2] The same ResNet model has been used in Allen-Zhu et al. (2018b) and Du et al. (2018). Here, we borrow notations from Allen-Zhu et al. (2018b).

converse side, if $\tau > L^{-0.5+c}$ for any positive constant $c$, the output norm grows at least polynomial with depth. The stability result has good correspondence with empirical observation. Moreover, it forms the basis for establishing the global convergence in Section 4.

## 3.1 FORWARD PROCESS IS BOUNDED FOR $\tau \leq 1/\Omega(\sqrt{L})$

We first give a non-asymptotic bound on the spectral norm of the forward process at initialization.

**Theorem 1.** *Suppose that $\overrightarrow{W}^{(0)}$, $A$ are randomly generated as in the initialization step, and $D_0, \ldots, D_L$ are diagonal matrices such that $\|D_l\|_2 \leq 1$ for all $l \in [L]$ and $D_l$ is deterministic given $\{W_a^{(0)} : a \leq l\}$. If $\tau \leq 1/\Omega(\sqrt{L})$, then there exists some small constant $c$ such that with probability at least $1 - L^2 \cdot \exp(-\Omega(mc^2))$ over the initialization randomness we have for any $b > a$,*

$$\left\| D_b \left( I + \tau W_b^{(0)} \right) D_{b-1} \cdots D_a \left( I + \tau W_a^{(0)} \right) \right\|_2 \leq 1 + c. \tag{2}$$

The above result is a bit surprising since for $\tau = 1/\Omega(\sqrt{L})$ a natural bound on the spectral norm $\|(I + \tau W_L^{(0)}) \cdots (I + \tau W_1^{(0)})\| \leq (1 + \frac{1}{\sqrt{L}})^L$ explodes. Here the intuition is that the cross-product term concentrates on the mean 0 because of the independent randomness of matrices $W_l^{(0)}$. Moreover, we note that to guarantee Theorem 1 holds for all training samples $[n]$, we take the union bound and the probability becomes $1 - nL^2 \cdot \exp(-\Omega(mc^2))$. Next we give a rigorous argument based on the martingale sequence.

*Proof Outline.* Suppose we have $\|h_{a-1}\| = 1$. Then we abuse notations $g_l = h_{l-1} + \tau W_l^{(0)} h_{l-1}$ and $h_l = D_l g_l$ for $a \leq l \leq b$, and we have

$$\|h_b\|^2 = \frac{\|h_b\|^2}{\|h_{b-1}\|^2} \cdots \frac{\|h_a\|^2}{\|h_{a-1}\|^2} \|h_{a-1}\|^2 \leq \frac{\|g_b\|^2}{\|h_{b-1}\|^2} \cdots \frac{\|g_a\|^2}{\|h_{a-1}\|^2} \|h_{a-1}\|^2.$$

Taking logarithm at both side, we have

$$\log \|h_b\|^2 \leq \sum_{l=a}^{b} \log \Delta_l, \qquad \text{where } \Delta_l := \frac{\|g_l\|^2}{\|h_{l-1}\|^2}.$$

If let $\tilde{h}_{l-1} := \frac{h_{l-1}}{\|h_{l-1}\|}$, then we obtain that

$$\log \Delta_l = \log \left( 1 + 2\tau \left\langle \tilde{h}_{l-1}, W_l^{(0)} \tilde{h}_{l-1} \right\rangle + \tau^2 \|W_l^{(0)} \tilde{h}_{l-1}\|^2 \right) \leq 2\tau \left\langle \tilde{h}_{l-1}, W_l^{(0)} \tilde{h}_{l-1} \right\rangle + \tau^2 \|W_l^{(0)} \tilde{h}_{l-1}\|^2,$$

where the inequality is because $\log(1 + x) < x$ for $x > -1$. Let $\xi_l := 2\tau \left\langle \tilde{h}_{l-1}, W_l^{(0)} \tilde{h}_{l-1} \right\rangle$ and $\zeta_l := \tau^2 \|W_l^{(0)} \tilde{h}_{l-1}\|^2$, for given $\tilde{h}_{l-1}$, $\xi_l \sim \mathcal{N}\left(0, \frac{8\tau^2}{m}\right)$, $\zeta_l \sim \frac{2\tau^2}{m} \chi_m^2$.

Without rigor, we could say $\sum_{l=a}^{b} \xi_l \sim \mathcal{N}\left(0, \frac{8(b-a)\tau^2}{m}\right)$ and $\sum_{l=a}^{b} \zeta_l \sim \frac{2(b-a)\tau^2}{m} \chi_m^2$. Hence we have $\sum_{l=a}^{b} \log \Delta_l \leq c$ with probability at least $1 - \exp(-\Omega(m)c^2)$. Taking $\varepsilon$-net argument, we can establish the spectral norm bound for all vector $h_{a-1}$. Let $a$ and $b$ vary from 1 to $L-1$ and taking the union bound gives the claim. □

We next show that the output norm at each layer is close to 1.

**Theorem 2.** *Suppose $\tau \leq 1/\Omega(\sqrt{L})$. There exists some small constant $c$ such that with probability at least $1 - O(nL) \cdot e^{-\Omega(m \cdot \min(c,c^2))}$ over the randomness of $A \in \mathbb{R}^{m \times p}$ and $\overrightarrow{W}^{(0)} \in (\mathbb{R}^{m \times m})^L$ the following holds*

$$\forall i \in [n], l \in \{0, 1, \ldots, L\} : \quad \left\| h_{i,l}^{(0)} \right\| \in [1 - c, 1 + c]. \tag{3}$$

The proof is relegated to Appendix C.1. We note that Theorem 2 is much stronger compared with the result in Allen-Zhu et al. (2018b) which is only showed for the case $\tau = 1/\Omega(L \log m)$. Moreover the above two lemmas also holds for $\overrightarrow{W}$ that is within the neighborhood of $\overrightarrow{W}^{(0)}$ and the result is presented in Appendix C.2.

In the sequel, $c$ is treated as a fixed constant, e.g. $c = 0.1$, which may be hidden in the $\Omega()$ or $O()$ notation.

## 3.2 BACKWARD PROCESS IS BOUNDED FOR $\tau \leq 1/\Omega(\sqrt{L})$

For ResNet, the gradient with respect to the parameter is computed through back-propagation, e.g., $\partial \boldsymbol{W}_l = \partial h_l \cdot h_{l-1}^T$, where $\partial \cdot$ represents the gradient of the objective with respect to $\cdot$. Therefore, the gradient upper bound is guaranteed if $h_l$ and $\partial h_l$ are bounded across layers and iterations. We next show the backward process is bounded for each individual sample.

**Theorem 3.** *With probability at least* $1 - (nL) \cdot \exp(-\Omega(m))$ *over the randomness of* $\overrightarrow{\boldsymbol{W}}^{(0)}, \boldsymbol{A}, \boldsymbol{B}$, *it satisfies for every* $l \in [L-1]$, *every* $i \in [n]$, *and every* $\overrightarrow{\boldsymbol{W}}$ *with* $\|\overrightarrow{\boldsymbol{W}} - \overrightarrow{\boldsymbol{W}}^{(0)}\|_2 \leq \omega$ *for* $\omega \in [0, 1]$,

$$\|\nabla_{\boldsymbol{W}_l} F_i(\overrightarrow{\boldsymbol{W}})\|_F^2 \leq O\left(\frac{F_i(\overrightarrow{\boldsymbol{W}})}{d} \times \tau^2 m\right), \qquad \|\nabla_{\boldsymbol{W}_L} F_i(\overrightarrow{\boldsymbol{W}})\|_F^2 \leq O\left(\frac{F_i(\overrightarrow{\boldsymbol{W}})}{d} \times m\right). \quad (4)$$

The full proof is relegated to Appendix D.1. Here we give an outline.

*Proof Outline.* The argument is based on the bounded forward/backward process at $\overrightarrow{\boldsymbol{W}}$ and the back-propagation formula. For each $i \in [n]$ and $l \in [L-1]$, i.e., the residual layers, we have

$$\|\nabla_{\boldsymbol{W}_l} F_i(\overrightarrow{\boldsymbol{W}})\|_F = \tau \left(\boldsymbol{D}_{i,l}(\boldsymbol{I} + \tau \boldsymbol{W}_{l+1})^T \cdots \boldsymbol{D}_{i,L-1} \boldsymbol{W}_L^T \boldsymbol{D}_{i,L} \boldsymbol{B}^T \left(\boldsymbol{B} h_{i,L} - y_i^*\right)\right) h_{i,l-1}^T$$

$$\leq O(\tau\sqrt{m/d})\sqrt{F_i(\overrightarrow{\boldsymbol{W}})},$$

where the last inequality is due to that the forward/backward process is bounded for all the $\overrightarrow{\boldsymbol{W}}$ such that $\|\overrightarrow{\boldsymbol{W}} - \overrightarrow{\boldsymbol{W}}^{(0)}\|_2 \leq \omega$ (Lemma 3 in Appendix). $\square$

This gradient upper bound indicates that the gradient of residual layers could be much ($\tau < 1$) smaller than the usual feedfoward layer.

## 3.3 A CONVERSE RESULT FOR $\tau > 1/\Omega(\sqrt{L})$

We have built the stability of the forward/backward process for $\tau \leq 1/\Omega(\sqrt{L})$. We next establish a converse result showing that if $\tau$ is slightly larger than $1/\Omega(\sqrt{L})$, the network output norm grows uncontrollably as the depth $L$ increases, which justifies the tightness of the value $\tau = 1/\Omega(\sqrt{L})$ for arbitrary $L$.

**Theorem 4.** *For the ResNet defined and initialized as in Section 2, if* $\tau \geq L^{-\frac{1}{2}+c}$, *then in expectation*

$$\boldsymbol{E}\|h_L\|^2 > L^{2c}. \quad (5)$$

*Proof.* The proof is relegated to the supplemental material in Appendix G. $\square$

This indicates the value of $\tau = 1/\Omega(\sqrt{L})$ is tight for characterizing the forward stability of learning ResNet. We note that the theoretical results in Section 3 hold for very mild condition i.e., high probabiltiy is obtained when $m > \Omega(\log(nL))$. In the next section, we will show that gradient descent is guaranteed to find the global minima for training ResNet if the network is properly over-parameterized when $\tau \leq 1/\Omega(\sqrt{L})$.

Up to now, we have provided a full characterization of the ResNet stability in terms of the value of $\tau$. Next, we take one step further towards bridging the theory to practice. We study whether this theoretical result has any practical guide. In practice, instead of using $\tau$, the normalization layers, e.g. batch normalization (BN) and layer normalization (LN), are used to control the forward/backward stability. One first guide is that ResNet can be effectively trained even without normalization layer if choosing $\tau = 1/\sqrt{L}$, as shown in the experiments in Section 5.

Our experiments demonstrate that the training performance of ResNet with $\tau$ is on par with the ResNet with BN for all depths $\{20, 32, 56, 110, 1202\}$ on CIFAR10 classification task. The test performance is a bit complicated: ResNets with $\tau$ drop $1 \sim 2$ points compared to ResNet with BN for depths $\{20, 32, 56, 110\}$, which may be attributed to the regularization effect of BN (Zhang et al.,

2019a); for depth 1202, ResNet with $\tau$ is slightly better than ResNet with BN. This motivates us to study the potential benefit of combining BN and $\tau$.

We observe that although the residual block output norm of ResNet with BN does not increase exponentially, it still increases as traversing through layers. We give an heuristic estimation on how the residual block output norm grows for ResNet with BN. Take the image classification task as an example, the input is normalized across channel and batch. Therefore, for $(\tilde{g}_l)_k = \mathrm{BN}\left((\boldsymbol{W}_l h_{l-1})_k\right)$, we assume that $\mathrm{Var}[(\tilde{g}_l)_k] = 1$ for all $k = 1, ..., m$ and $l = 0, 1, ..., L-1$. We further assume the independence of each $(\tilde{g}_l)_k$. Then, we have the following estimation.

**Claim 1.** *The output for ResNet with BN grows with* $\boldsymbol{E}\|\boldsymbol{h}_L\|_F^2 \approx nmL$, *where* $\boldsymbol{h}_L = [h_{L1}, ...., h_{Ln}]$.

The calculation can be adapted from the proof of Theorem 4. This indicates that the output norm of ResNet with BN grows roughly at the rate $\sqrt{L}$. We propose that adding $\tau = 1/\sqrt{L}$ on top of BN can keep the output norm roughly constant across layers, which produces considerable performance gain especially for deep ResNet.

## 4 GRADIENT DESCENT CONVERGES TO GLOBAL MINIMA FOR LEARNING OVER-PARAMETERIZED RESNET

In this section, we establish that gradient descent can converge to global minima for learning over-parameterized ResNet with $\tau \leq 1/\Omega(\sqrt{L})$. Compared to the recent work (Allen-Zhu et al., 2018b), our result enlarges the region of $\tau$ that admits the global convergence of gradient descent. Moreover, our result also theoretically justifies the advantage of ResNet over vanilla feedforward network in terms of facilitating the convergence of gradient descent. Before stating the theorem, we introduce a common assumption on the training data (Allen-Zhu et al., 2018b; Zou and Gu, 2019; Oymak and Soltanolkotabi, 2019).

**Assumption 1** (training data). *For any $x_i$, it holds that $\|x_i\| = 1$ and $(x_i)_p = 1/\sqrt{2}$. For every pair $i, j \in [n]$, we assume $\|x_i - x_j\| \geq \delta$.*

We further assume that the loss function $\ell(\cdot, \cdot)$ is quadratic and the objective function for individual sample becomes $F_i(\overrightarrow{\boldsymbol{W}}) := \frac{1}{2}\|\boldsymbol{B}h_{i,L} - y_i^*\|^2$.

**Theorem 5.** *Suppose that the ResNet is defined as in Section 2 with $\tau \leq 1/\Omega(\sqrt{L}\log m)$ and training data satisfy Assumption 1. If the network width $m \geq \Omega(n^8 L^7 \delta^{-4} d \log^2 m)$, then with probability at least $1 - \exp(-\Omega(\log^2 m))$, gradient descent with learning rate $\eta = \Theta(\frac{d}{nm})$ finds a point $F(\overrightarrow{\boldsymbol{W}}) \leq \varepsilon$ in $T = \Omega(n^2 \delta^{-1} \log \frac{n \log^2 m}{\varepsilon})$ iterations.*

*Proof.* The proof is deferred to Appendix F. □

This theorem establishes the linear convergence of gradient descent for learning ResNet with $\tau \leq 1/\Omega(\sqrt{L}\log m)$. Compared with Allen-Zhu et al. (2018b); Du et al. (2018), our result significantly enlarges the range of $\tau$ (from $\tau \leq 1/\Omega(L\log m)$ to $\tau \leq 1/\Omega(\sqrt{L}\log m)$) that can guarantee the global convergence of training over-parameterized ResNet.

Moreover, we argue that even for $\tau = 1/\Omega(\sqrt{L}\log m)$, the depth dependence for learning over-parameterized ResNet is smaller than that for learning feedforward network, which is also better than the bound for ResNet given in previous work (Allen-Zhu et al., 2018b). This theoretically justifies the advantage of ResNet over vanilla feedforward network in terms of facilitating the convergence of gradient descent. For the case of $\tau \leq 1/\Omega(L\log m)$, the depth dependence might be further reduced at the cost of adding the order on $n$. We believe that the depth dependence of ResNet is due to the limit of bounding techniques when handling nonsmooth activation. Better bounding technique or smooth activation function may help remove such dependence. We leave this for future study. For the case of $\tau > 1/\Omega(\sqrt{L})$, the global convergence cannot be guaranteed because of the instability of the forward process for extremely large depth as shown in Section 3.3.

### 4.1 PROOF SKETCH AND MAIN CHALLENGES

Though the objective is nonconvex and nonsmooth from the first sight, it admits good optimization property, which can guarantee the convergence of gradient descent, at the initialization and along the optimization path. Specifically, we establish three properties in the neighborhood of initialization: *gradient is upper bounded*, *gradient is lower bounded*, i.e., the gradient is large when the objective is large, and the objective satisfies certain *smooth property*.

The gradient upper and lower bounds for each individual sample have been proved based on *the stability of forward/backward process* at the initialization stage and after small perturbation in Section 3.2. Then following Allen-Zhu et al. (2018b); Zou and Gu (2019), we argue the sum of individual gradients also being large. The **gradient lower bound** is presented as Theorem 7 in Appendix D.2.

The next challenge is to establish the **objective smoothness**. We present an informal semi-smoothness[3] result. Interestingly, from this result, we can illustrate the advantage of learning ResNet: the convergence of learning ResNet only depends on the network depth weakly, echoing the empirical evidence that deep ResNet is much easier to train than deep feedforward network.

**Theorem 6** (Informal semi-smoothness result). *Let $\omega < 1$ and $\tau^2 L \leq 1$. With high probability, we have for every $\overset{\breve{}}{\overrightarrow{W}} \in (\mathbb{R}^{m \times m})^L$ with $\|\breve{W}_L - W_L^{(0)}\|_2 \leq \omega$ and $\|\breve{W}_l - W_l^{(0)}\|_2 \leq \tau\omega$ for $l \in [L-1]$, and for every $\overrightarrow{W}' \in (\mathbb{R}^{m \times m})^L$ with $\|W_L'\|_2 \leq \omega$ and $\|W_l'\|_2 \leq \tau\omega$ for $l \in [L-1]$, we have*

$$F(\overset{\breve{}}{\overrightarrow{W}} + \overrightarrow{W}') \leq F(\overset{\breve{}}{\overrightarrow{W}}) + \langle \nabla F(\overset{\breve{}}{\overrightarrow{W}}), \overrightarrow{W}' \rangle + O\left(\frac{nm}{d}\right)\|\overrightarrow{W}'\|_F^2 + O\left(\sqrt{\frac{mnL\omega^{2/3}}{d}}(\tau L)^{4/3}\right)\|\overrightarrow{W}'\|_F\sqrt{F(\overset{\breve{}}{\overrightarrow{W}})}.$$

We note that apart from the second-order term (the third term on the right hand side) in classical smoothness, the semi-smoothness has a first-order term (the last term on the right hand side). One can see that as $m$ becomes large in the over-parameterization regime the effect of the first-order term becomes small comparing to the second-order term. Interestingly, if one replaces $W_L'$ and $W_l'$ with the gradient upper bounds in Theorem 3, only the first-order term depends on $L$ while the second-order term, which is dominant when $m$ is large, is depth-independent. We note that the first-order term is the only source where the depth dependence in Theorem 5 comes from, which renders the depth dependence is weak for learning ResNet when $m$ is large, in contrast with the case of learning deep feedforward network (Allen-Zhu et al., 2018b).

## 5 EMPIRICAL STUDY

In this section, we present some experiments to verify our theory. We first demonstrate that $\tau = 1/\sqrt{L}$ is a sharp value in determining the trainability of deep ResNet. We then show that practical ResNet with $\tau$ can be efficiently trained even without normalization layer. We finally show for ResNet with normalization, that adding $\tau$ also achieve considerable performance gain for both CIFAR and ImageNet tasks.

### 5.1 THEORETICAL VERIFICATION

We train feedforward fully-connected neural networks and ResNets with different values of $\tau$, and compare their convergence behaviors. The feedforward model adopts the same architecture and initialization scheme as the ResNet model except the skip connection (see Section 2). The models are generated with width $m = 128$ and depth $L \in \{3, 10, 30, 100, 500, 1000\}$ and . We conduct experiments with $\tau = \frac{1}{L}, \frac{1}{L^{0.5}}, \frac{1}{L^{0.25}}$ and show how it affects the training performance. We use MNIST dataset (LeCun et al., 1998) and do the classification task. We train the model with SGD [4] and the size of minibatch is 256. The learning rate $lr$ is set as $lr = 0.001$ without heavily tuning.

**Experiment results.** We plot the training curves of feedforward network and ResNet with varying depths and widths in Figure 1. We see that both $\tau = \frac{1}{L}$ and $\tau = \frac{1}{L^{0.5}}$ are able to train very deep ResNets successfully. However, when $\tau = \frac{1}{L^{0.25}}$, the training loss explodes for models with depth 30

---

[3]The smoothness is compromised from the usual sense because of the non-smoothness of ReLU.

[4]GD exhibits the same phenomenon. We use SGD due to the expensive per-iteration cost of GD.

and more. This indicates that the bound $\tau = \frac{1}{\sqrt{L}}$ is sharp for learning ResNet with arbitrary depth. Moreover the convergence of ResNets with $\tau = \frac{1}{L}$ and $\tau = \frac{1}{L^{0.5}}$ do not depend on the depth much while training feedforward network becomes harder as the depth increases, which verifies our theory of weak dependence on depth for learning ResNet.

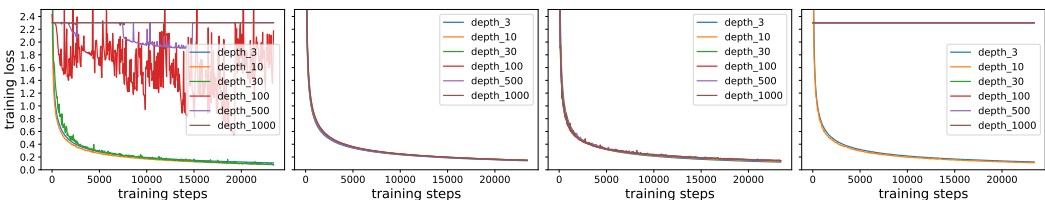

Figure 1: Training curves for feedforward network and ResNets with $\tau = \frac{1}{L}, \frac{1}{L^{0.5}}$ and $\frac{1}{L^{0.25}}$ (from left to right).

## 5.2 LEARN RESNET WITH $\tau$

We use more experiments to demonstrate that **ResNet can be trained efficiently with $\tau$ even when there is not normalization layer.**

In this section we conduct experiments on CIFAR10/100 datasets. We use the ResNet models (He et al., 2016) with the same hyperparameters but remove all the normalization layers. Motivated by Zhang et al. (2019a), we add learnable scalar bias at the input of each convolution layer. Moreover, we treat $\tau$ as a learnable parameter with initialization $1/\sqrt{L}$. Following Zhang et al. (2019a), the learning rate of scalar parameters is divided by 10. All blocks share the same $\tau$ and we use the averaged gradient to update $\tau$. Figure 2 shows the training/validation curves.

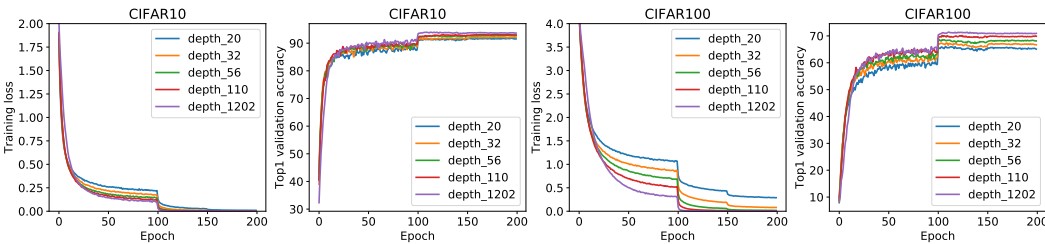

Figure 2: Experiment results on CIFAR10/100 with $\tau$ initialized as $\frac{1}{\sqrt{L}}$. All normalization layers are removed.

With $\tau$ initialized as $\frac{1}{\sqrt{L}}$, we can easily train ResNet without batch normalization (BN) even for very deep networks. We note that Zhang et al. (2019a) has also proposed a way to train ResNet without normalization layers by doing specific initialization strategy (scaling down the first weight matrix and zeroing the last weight matrix inside all residual blocks). We compare the performances of our $\tau$ ResNet strategy and the *Fixup* scheme in Table 1. From Table 1, we see that scaling down the output of residual block has better performance than scaling down the initialization. Moreover, the Fixup scheme fails to converge 2 out of 5 runs for training ResNet1202 while there is no failure case for our $\tau$ ResNet. This indicates that the Fixup scheme could be unstable for extremely deep ResNet. The possible reason is that $\tau$ can stabilize both the forward pass and the backward pass while scaling down initialization only stabilizes the forward pass.

We also conduct experiments on *Transformer (Vaswani et al., 2017)* for machine translation task and compare our $\tau$ and the Fixup scheme Zhang et al. (2019a). Transformer uses multiple residual connections in its basic building block. Therefore multiplying $\tau$ after each residual connection can also stabilize its training process. The results is relegated to Appendix H.

| Dataset | Depth | ResNet + FixupInit | ResNet + $\tau$ |
|---------|-------|--------------------|-----------------|
| | 20 | 8.72($\pm$0.26) | **8.39($\pm$0.11)** |
| | 32 | 7.99($\pm$0.24) | **7.68($\pm$0.10)** |
| CIFAR 10 | 56 | 7.45($\pm$0.37) | **7.07($\pm$0.16)** |
| | 110 | 7.24($\pm$0.12) | **6.52($\pm$0.20)** |
| | 1202 | 7.83($\pm$0.18) | **6.08($\pm$0.21)** |

Table 1: Top1 validation error on CIFAR10. Numbers are average of 5 runs except ResNet1202 + Fixup (standard deviations are given inside the bracket).

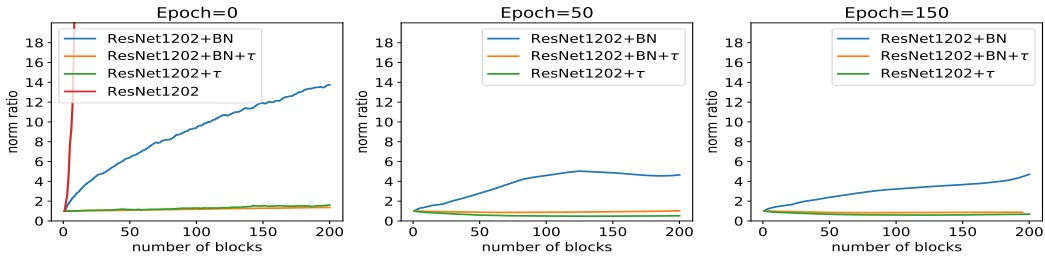

Figure 3: Output norm of ResNet1202 over layers at different epochs. X axis is the index of block. Y axis is the output norm ratio compared to the first block.

## 5.3 ADD $\tau$ ON TOP OF NORMALIZATION

In this section, we investigate the property of ResNet model with normalization layer and empirically demonstrate that adding a $\tau$ on top of normalization can achieve even better performance.

We first illustrate how the output norm of each residual block grows for ResNet1202 with BN (He et al., 2016; Ioffe and Szegedy, 2015) in Figure 3. We see that at epoch 0 (initialization stage), the output norm grows almost at the rate $\sqrt{l}$ as we estimate in Claim 1. After training, the estimation in Claim 1 is not as accurate as the initialization because the independence assumption does not hold after training. Nonetheless, by adding a $\tau$ on top of BN, the output norm keeps roughly constant across the forward layers and across the training epochs.

We next verify that **adding $\tau$ can further improve the performance over original models.**

*Experiments on image classification.* We use the standard classification datasets: CIFAR10/100 and ImageNet. Our models and hyperparameters are the same as in He et al. (2016). The only modification is multiplying a fixed $\tau = \frac{1}{\sqrt{L}}$ at the output of each residual block (right before the residual addition). For ResNet1202, we do not use small learning rate to warm up the training. The validation errors on CIFAR10/100 are illustrated in Figure 4, where all numbers are averaged over five runs. We note that the benefit of $\tau$ becomes larger when the network is deeper. Without $\tau$, the performance of ResNet1202 on CIFAR10 is worse than ResNet110 as shown in He et al. (2016). As depth increases, the norm grow and imbalance over layers hurts the performance, which covers up the benefit of adding layers. As shown in Figure 4, ResNets+BN+$\tau$ keep the output norm roughly constant and make good use of the network depth.

The ResNet model for ImageNet has different numbers of residual blocks in each stage, and we choose $L$ to represent the largest number of blocks over all stages. We choose $\tau = \frac{2}{\sqrt{L}}$ for ImageNet dataset. All models are trained for 200 epochs with learning rate divided by 10 every 60 epochs. The other hyperparameters are the same as in He et al. (2016).

Table 2 shows the results on ImageNet. We can see that by just adding a $\tau$ on top of BN we can achieve considerable performance gain. We note that a trick in Goyal et al. (2017) that initializes the scaling factor $\gamma$ of the last BN of each residual block to 0, shares the same spirit as a small $\tau$ here, but does not have principled justification.

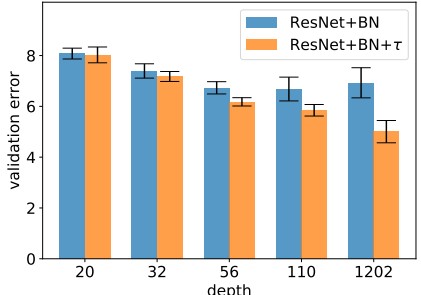 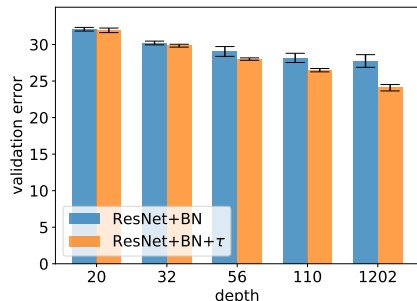

Figure 4: Top1 validation error on CIFAR10/CIFAR100 dataset. The benefit of $\tau$ becomes larger as network goes deeper.

| Model | Method | Validation Error (Top1) |
|---|---|---|
| ResNet50 | +BN | 23.6 |
| | +BN+$\tau$ | **23.0** |
| ResNet101 | +BN | 22.0 |
| | +BN+$\tau$ | **21.4** |
| ResNet152 | +BN | 21.7 |
| | +BN+$\tau$ | **20.9** |

Table 2: Validation error on ImageNet dataset.

## 6 CONCLUSION

In this paper, we provide a non-asymptotic analysis on the forward/backward stability for ResNet, which unveils that $\tau = 1/\sqrt{L}$ is a sharp value in terms of characterizing the stability. Furthermore, when the network is properly over-parameterized, we show that gradient descent finds global minima for training ResNet with $\tau \leq 1/\Omega(\sqrt{L}\log m)$ which greatly improves over previous work of $\tau \leq 1/\Omega(L\log m)$. We also bridge theoretical understanding and practical guide of ResNet structure. We empirically verify the efficacy of the suggestion $\tau = 1/\sqrt{L}$ for ResNet with/without batch normalization. One interesting future direction is to bypass the semi-smooth argument and give a sharp dependence on the depth and the number of training samples.

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

## A  USEFUL LEMMAS

First we list several useful bounds on Gaussian distribution.

**Lemma 1.** *Suppose $X \sim \mathcal{N}(0, \sigma^2)$, then*

$$\mathbb{P}\{|X| \leq x\} \geq 1 - \exp\left(-\frac{x^2}{2\sigma^2}\right), \tag{6}$$

$$\mathbb{P}\{|X| \leq x\} \leq \sqrt{\frac{2}{\pi}} \frac{x}{\sigma}. \tag{7}$$

Another bound is on the spectral norm of random matrix (Vershynin, 2012, Corollary 5.35).

**Lemma 2.** *Let $A \in \mathbb{R}^{N \times n}$, and entries of $A$ are independent standard Gaussian random variables. Then for every $t \geq 0$, with probability at least $1 - \exp(-t^2/2)$ one has*

$$s_{\max}(A) \leq \sqrt{N} + \sqrt{n} + t, \tag{8}$$

*where $s_{\max}(A)$ are the largest singular value of $A$.*

## B  SPECTRAL NORM BOUND AT INITIALIZATION

Next we present a spectral norm bound related to the forward process of ResNet with $\tau$.

**Theorem 1.** *Suppose that $\overrightarrow{W}^{(0)}$, $A$ are randomly generated as in the initialization step, and $D_0, \ldots, D_L$ are diagonal matrices such that $\|D_l\|_2 \leq 1$ for all $l \in [L]$ and $D_l$ is deterministic given $\{W_a^{(0)} : a \leq l\}$. If $\tau \leq 1/\Omega(\sqrt{L})$, then there exists some small constant $c$ such that with probability at least $1 - L^2 \cdot \exp(-\Omega(mc^2))$ over the initialization randomness we have for any $b > a$,*

$$\left\| D_b \left( I + \tau W_b^{(0)} \right) D_{b-1} \cdots D_a \left( I + \tau W_a^{(0)} \right) \right\|_2 \leq 1 + c. \tag{2}$$

*Proof.* We first show for any vector $h_{a-1}$ with $\|h_{a-1}\| = 1$, we have $\|h_b\| \leq 1 + c$ with high probability, where

$$h_b = D_b(I + \tau W_b^{(0)})D_{b-1} \cdots D_a(I + \tau W_a^{(0)})h_{a-1}. \tag{9}$$

Using notations $g_l = h_{l-1} + \tau W_l h_{l-1}$ introduced in Section 2, we have $\|g_l\| \geq \|h_l\|$. Thus we have the following

$$\|h_b\|^2 = \frac{\|h_b\|^2}{\|h_{b-1}\|^2} \cdots \frac{\|h_a\|^2}{\|h_{a-1}\|^2} \|h_{a-1}\|^2 \leq \frac{\|g_b\|^2}{\|h_{b-1}\|^2} \cdots \frac{\|g_a\|^2}{\|h_{a-1}\|^2} \|h_{a-1}\|^2.$$

Taking logarithm at both side, we have

$$\log \|h_b\|^2 \leq \sum_{l=a}^{b} \log \Delta_l, \qquad \text{where } \Delta_l := \frac{\|g_l\|^2}{\|h_{l-1}\|^2}. \tag{10}$$

If let $\tilde{h}_{l-1} := \frac{h_{l-1}}{\|h_{l-1}\|}$, then we obtain that

$$\log \Delta_l = \log \left( 1 + 2\tau \left\langle \tilde{h}_{l-1}, W_l^{(0)} \tilde{h}_{l-1} \right\rangle + \tau^2 \|W_l^{(0)} \tilde{h}_{l-1}\|^2 \right)$$

$$\leq 2\tau \left\langle \tilde{h}_{l-1}, W_l^{(0)} \tilde{h}_{l-1} \right\rangle + \tau^2 \|W_l^{(0)} \tilde{h}_{l-1}\|^2,$$

where the inequality is due to the fact $\log(1 + x) \leq x$ for all $x > -1$. Let $\xi_l := 2\tau \left\langle \tilde{h}_{l-1}, W_l^{(0)} \tilde{h}_{l-1} \right\rangle$ and $\zeta_l := \tau^2 \|W_l^{(0)} \tilde{h}_{l-1}\|^2$, then given $h_{l-1}$ we have $\xi_l \sim \mathcal{N}\left(0, \frac{8\tau^2}{m}\right)$, $\zeta_l \sim \frac{2\tau^2}{m} \chi_m^2$.

We see that

$$\mathbb{P}\left(\sum_{l=a}^{b} \log \Delta_l \geq c\right) \leq \mathbb{P}\left(\sum_{l=a}^{b} \xi_l \geq \frac{c}{2}\right) + \mathbb{P}\left(\sum_{l=a}^{b} \zeta_l \geq \frac{c}{2}\right). \tag{11}$$

Next we bound terms on the right hand side one by one. For the first term we have

$$\mathbb{P}\left(\sum_{l=a}^{b}\xi_l \geq \frac{c}{2}\right) = \mathbb{P}\left(\exp\left(\lambda\sum_{l=a}^{b}\xi_l\right) \geq \exp\left(\frac{\lambda c}{2}\right)\right) \leq \mathbb{E}\left[\exp\left(\lambda\sum_{l=a}^{b}\xi_l - \frac{\lambda c}{2}\right)\right], \quad (12)$$

where $\lambda$ is any positive number and the last inequality uses the Markov's inequality. Moreover,

$$\mathbb{E}\left[\exp\left(\lambda\sum_{l=a}^{b}\xi_l\right)\right] = \mathbb{E}\left[\exp\left(\lambda\sum_{l=a}^{b-1}\xi_l\right)\mathbb{E}\left[\exp\left(\lambda\xi_b\right)\Big|\mathcal{F}_{b-1}\right]\right]$$

$$= \exp\left(\frac{8\tau^2\lambda^2}{m}\right)\mathbb{E}\left[\exp\left(\lambda\sum_{l=a}^{b-1}\xi_l\right)\right] \quad (13)$$

$$= \cdots = \exp\left(\frac{8\tau^2\lambda^2(b-a+1)}{m}\right)$$

Hence we obtain

$$\mathbb{P}\left(\sum_{l=a}^{b}\xi_l \geq \frac{c}{2}\right) \leq \exp\left(-\frac{mc^2}{128\tau^2(b-a+1)}\right) = \exp\left(-\Omega\left(\frac{mc^2}{\tau^2(b-a+1)}\right)\right) \quad (14)$$

by choosing $\lambda = \frac{mc}{32\tau^2 l}$ and $\tau \leq 1/\Omega(\sqrt{L})$. Due to the symmetry of $\sum_{l=a}^{b}\xi_l$, the conclusion can be generalized to the quantity $|\sum_{l=a}^{b}\xi_l|$.

Then, for the second term, we first present a concentration inequality for the general $\chi_m^2$ distribution $X$ (Laurent and Massart, 2000, Lemma 1)

$$\mathbb{P}\left(|X - m| \geq u\right) \leq e^{-\frac{u^2}{4m}}. \quad (15)$$

Then for $\sum_{l=a}^{b}\zeta_l$, by applying the above concentration inequality and Jensen's inequality, we have

$$\mathbb{P}\left(\sum_{l=a}^{b}\zeta_l \geq \frac{c}{2}\right) = \mathbb{E}\left[\mathbb{P}\left(\sum_{l=a}^{b}\zeta_l \geq \frac{c}{2}\Big|\mathcal{F}_{b-1}\right)\right]$$

$$\leq \mathbb{E}\left[\mathbb{P}\left(\left|\zeta_b + \sum_{l=a}^{b-1}\zeta_l - 2\tau^2(b-a+1+1)\right| \geq \frac{c}{2} - 2\tau^2(b-a+1)\Big|\mathcal{F}_{b-1}\right)\right]$$

$$\leq \mathbb{E}\left[\mathbb{P}\left(\left|\zeta_b - 2\tau^2\right| \geq \frac{c}{2} - (4L-2)\tau^2 - \sum_{k=a}^{b-1}\zeta_k\Big|\mathcal{F}_{b-1}\right)\right]$$

$$= \mathbb{E}\left[\mathbb{P}\left(\left|\frac{m}{2\tau^2}\zeta_b - m\right| \geq \frac{m}{2\tau^2}\left(\frac{c}{2} - (4L-2)\tau^2 - \sum_{k=a}^{b-1}\zeta_k\right)\Big|\mathcal{F}_{b-1}\right)\right]$$

$$\leq \mathbb{E}\left[\exp\left(-\frac{m}{16\tau^4}\left(\frac{\delta}{2} - (4L-2)\tau^2 - \sum_{k=a}^{b-1}\zeta_k\right)^2\right)\right]$$

$$\leq \mathbb{E}\left[\exp\left(-\frac{m}{16\tau^4}\left(\frac{c^2}{4} + \Omega(L^2)\tau^4 - \Omega(L)\tau^2\right)\right)\right]$$

$$= \exp\left(-\Omega\left(\frac{mc^2}{\tau^4}\right)\right) \quad (16)$$

Combining equation 16, equation 14 and the condition $\tau \leq \frac{1}{\Omega(\sqrt{L})}$, we obtain $\|h_b\| \leq 1 + c$ with probability at least $1 - \exp(-\Omega(\frac{mc^2}{\tau^2 L}))$, where we use approximation $\log(1+c) \approx c$ for small $c$ to simplify the expression. Taking $\epsilon$-net over all $m$-dimensional vectors of $h_{a-1}$, with probability $1 - \exp(-\Omega(mc^2))$ the inequality 2 holds for a fixed $a$ and $b$ with $1 \leq a \leq b < L$. Taking a union bound over $a$ and $b$, the conclusion is proved.

The $\epsilon$-net argument is as follows. We have proved for a given unit vector $h_{a-1}$, $\|h_b\| > 1 + c$ with probability at most $\exp(-\Omega(\frac{mc^2}{\tau^2 L}))$ for a small constant $c$. Let $\mathcal{N}_\epsilon$ be an $\varepsilon$-net over the unit ball in $\mathbb{R}^m$ with $\epsilon = 1/11$, then we have the cardinality $|\mathcal{N}_\epsilon| \leq (1 + 2/\epsilon)^m < (23)^m$. Taking the union bound over all vectors $h_{a-1}$ in the net $\mathcal{N}_\epsilon$, we obtain

$$\mathbb{P}\left\{\max_{h_{a-1}\in\mathcal{N}_\epsilon} \|h_b\| > 1 + c\right\} \leq (1 + 2/\epsilon)^m \cdot \exp\left(-\Omega\left(\frac{mc^2}{\tau^2 L}\right)\right)$$
$$= \exp\left(-m\left(\Omega(\frac{mc^2}{\tau^2 L}) - \log 23\right)\right).$$

By choosing the coefficient of $\tau$ appropriately, we can make $\Omega(\frac{mc^2}{\tau^2 L}) > \log 23$ and then the right hand side can be written as $\exp\left(-m\left(-\Omega(\frac{mc^2}{\tau^2 L})\right)\right)$. Based on the result (Vershynin, 2012, Lemma 5.3), we have

$$\left\|D_b\left(I + \tau W_b^{(0)}\right) D_{b-1} \cdots D_a\left(I + \tau W_a^{(0)}\right)\right\|_2 \leq (1-\epsilon)^{-1} \max_{h_{a-1}\in\mathcal{N}_\epsilon} \|h_b\|_2 = 1.1 \max_{h_{a-1}\in\mathcal{N}_\epsilon} \|h_b\|_2.$$

We complete the $\epsilon$-net argument for the spectral norm bound by introducing a new constant $c$.

$\square$

## C   BOUNDED FORWARD/BACKWARD PROCESS

### C.1   PROOF AT INITIALIZATION

**Theorem 2.** *Suppose $\tau \leq 1/\Omega(\sqrt{L})$. There exists some small constant $c$ such that with probability at least $1 - O(nL) \cdot e^{-\Omega(m \cdot \min(c, c^2))}$ over the randomness of $A \in \mathbb{R}^{m \times p}$ and $\overrightarrow{W}^{(0)} \in (\mathbb{R}^{m \times m})^L$ the following holds*

$$\forall i \in [n], l \in \{0, 1, \ldots, L\}: \ \left\|h_{i,l}^{(0)}\right\| \in [1 - c, 1 + c]. \tag{3}$$

*Proof.* We ignore the subscript $(0)$ for simplicity. The upper bound of $\|h_{i,l}\|$ can be easily achieved by the proof of Theorem 1. Now, we give the lower bound of $\|h_{i,l}\|$. First we have

$$\|h_{i,l}\| = \|h_{i,0}\| \frac{\|h_{i,1}\|}{\|h_{i,0}\|} \cdots \frac{\|h_{i,l}\|}{\|h_{i,l-1}\|}. \tag{17}$$

Then we see

$$\log \|h_{i,l}\|^2 = \log \|h_{i,0}\|^2 + \sum_{a=1}^{l} \log \frac{\|h_{i,a}\|^2}{\|h_{i,a-1}\|^2}$$
$$= \log \|h_{i,0}\|^2 + \sum_{a=1}^{l} \log\left(1 + \frac{\|h_{i,a}\|^2 - \|h_{i,a-1}\|^2}{\|h_{i,a-1}\|^2}\right) \tag{18}$$
$$\geq \log \|h_{i,0}\|^2 + \sum_{a=1}^{l} \left(\Delta_a - \Delta_a^2\right),$$

where $\Delta_a := \frac{\|h_{i,a}\|^2 - \|h_{i,a-1}\|^2}{\|h_{i,a-1}\|^2}$ and the last inequality uses the relation $\log(1 + x) \geq x - x^2$. We next give a lower bound on $\Delta_a$. Let $S$ be the set $\{k : k \in [m] \text{ and } (h_{i,a-1})_k + (W_a h_{i,a-1})_k > 0\}$.

We have that

$$
\begin{aligned}
\Delta_a &= \frac{1}{\|h_{i,a-1}\|^2} \sum_{k \in S} \left[ (h_{i,a-1})_k^2 + 2\tau(h_{i,a-1})_k (\boldsymbol{W}_a h_{i,a-1})_k + (\tau \boldsymbol{W}_a h_{i,a-1})_k^2 \right] - \frac{1}{\|h_{i,a-1}\|^2} \sum_{k=1}^{m} (h_{i,a-1})_k^2 \\
&= -\frac{1}{\|h_{i,a-1}\|^2} \sum_{k \notin S} (h_{i,a-1})_k^2 + \frac{1}{\|h_{i,a-1}\|^2} \sum_{k \in S} \tau^2 (\boldsymbol{W}_a h_{i,a-1})_k^2 + \frac{2}{\|h_{i,a-1}\|^2} \sum_{k \in S} \tau (h_{i,a-1})_k (\boldsymbol{W}_a h_{i,a-1})_k \\
&\geq -\frac{1}{\|h_{i,a-1}\|^2} \sum_{k=1}^{m} (\tau \boldsymbol{W}_a h_{i,a-1})^2 + \frac{2}{\|h_{i,a-1}\|^2} \tau \sum_{k=1}^{m} (h_{i,a-1})_k (\boldsymbol{W}_a h_{i,a-1})_k \\
&= -\frac{\|\tau \boldsymbol{W}_a h_{i,a-1}\|^2}{\|h_{i,a-1}\|^2} + \frac{2\tau \langle h_{i,a-1}, \boldsymbol{W}_a h_{i,a-1} \rangle}{\|h_{i,a-1}\|^2},
\end{aligned}
\tag{19}
$$

where the inequality is due to the fact that for $k \notin S$, $|(h_{i,a-1})_k| < |(\tau \boldsymbol{W}_a h_{i,a-1})_k|$ and $(h_{i,a-1})_k (\boldsymbol{W}_a h_{i,a-1})_k \leq 0$. We let $\xi_a := \frac{2\tau \langle h_{i,a-1}, \boldsymbol{W}_a h_{i,a-1} \rangle}{\|h_{i,a-1}\|^2}$ and $\zeta_a := \frac{\|\tau \boldsymbol{W}_a h_{i,a-1}\|^2}{\|h_{i,a-1}\|^2}$, then $\Delta_a \geq \xi_a - \zeta_a$. We note that given $h_{i,a-1}$, $\xi_a \sim \mathcal{N}\left(0, \frac{8\tau^2}{m}\right)$ and $\zeta_a \sim \frac{2\tau^2}{m} \chi_m^2$. Due to equation 14 and equation 16, we have

$$
\begin{aligned}
\mathbb{P}\left( \left| \sum_{l=a}^{b} \xi_l \right| \geq \frac{c}{2} \right) &\leq 2 \exp\left( -\frac{mc^2}{128\tau^2(b-a+1)} \right) \\
\mathbb{P}\left( \sum_{l=a}^{b} \zeta_l \geq \frac{c}{2} \right) &\leq \exp\left( -\Omega\left( \frac{mc^2}{\tau^4} \right) \right)
\end{aligned}
\tag{20}
$$

Then for any $c > 0$, and $\tau \leq \frac{1}{\Omega(\sqrt{L})}$, we have

$$
\begin{aligned}
\mathbb{P}\left( \sum_{a=1}^{l} \Delta_a \leq -c \right) &= \mathbb{P}\left( \sum_{a=1}^{l} \Delta_a \leq -c, \sum_{a=1}^{l} \xi_a \geq -\frac{c}{2} \right) + \mathbb{P}\left( \sum_{a=1}^{l} \Delta_a \leq -c, \sum_{a=1}^{l} \xi_a \leq -\frac{c}{2} \right) \\
&\leq \mathbb{P}\left( \sum_{a=1}^{l} \zeta_a \geq \frac{c}{2} \right) + \mathbb{P}\left( \sum_{a=1}^{l} \xi_a \leq -\frac{c}{2} \right) = e^{-\Omega(c^2 m)}.
\end{aligned}
\tag{21}
$$

We can derive a similar result that $\mathbb{P}\left( \sum_{a=1}^{l} \Delta_a \geq c \right) \leq e^{-\Omega(c^2 m)}$. Let $a = b$ in equation 20, we obtain that for a single $\Delta_a$,

$$
\mathbb{P}\left( |\Delta_a| \geq c \right) \leq 2e^{-\Omega(Lmc^2)}
\tag{22}
$$

In addition, we see that

$$
\mathbb{P}\left( \sum_{a=1}^{l} \Delta_a^2 \geq c \right) \leq \sum_{a=1}^{l} \mathbb{P}\left( \Delta_a^2 \geq \frac{c}{l} \right) = \sum_{a=1}^{l} \mathbb{P}\left( |\Delta_a| \geq \sqrt{\frac{c}{l}} \right) \leq 2le^{-\Omega(-mc)}.
\tag{23}
$$

Thus, similar to the equation 21, we obtain

$$
\mathbb{P}\left( \sum_{a=1}^{l} \Delta_a - \Delta_a^2 \leq -c \right) \leq 2Le^{-\Omega\left(-m \min\{c, c^2\}\right)},
\tag{24}
$$

which results in

$$
\mathbb{P}\left( \log \|h_{i,l}\|^2 \leq -c \right) \leq \mathbb{P}\left( \log \|h_{i,0}\|^2 + \sum_{a=1}^{l} \left( \Delta_a - \Delta_a^2 \right) \leq -c \right) \leq 2Le^{-\Omega\left(\min\{c, c^2\}m\right)}.
\tag{25}
$$

Then we get the conclusion.

$\square$

## C.2 Lemmas and Proofs after Perturbation

We use $\overrightarrow{\boldsymbol{W}}^{(0)}$ to denote the weight matrices at initialization and use $\overrightarrow{\boldsymbol{W}}'$ to denote the perturbation matrices. Let $\overrightarrow{\boldsymbol{W}} = \overrightarrow{\boldsymbol{W}}^{(0)} + \overrightarrow{\boldsymbol{W}}'$. We define $h_{i,l}^{(0)} = \phi((\boldsymbol{I} + \tau \boldsymbol{W}_l^{(0)})h_{i,l-1}^{(0)})$ and $h_{i,l} = \phi((\boldsymbol{I} + \tau \boldsymbol{W}_l)h_{i,l-1})$ for $l \in [L-1]$, and $h_{i,L}^{(0)} = \phi(\boldsymbol{W}_L^{(0)} h_{i,L-1}^{(0)})$ and $h_{i,L} = \phi(\boldsymbol{W}_L h_{i,L-1})$. Furthermore, let $h_{i,l}' := h_{i,l} - h_{i,l}^{(0)}$ and $\boldsymbol{D}_{i,l}' := \boldsymbol{D}_{i,l} - \boldsymbol{D}_{i,l}^{(0)}$. Then the spectral norm bound after perturbation is as follows.

**Lemma 3.** *Suppose that $\overrightarrow{\boldsymbol{W}}^{(0)}$, $\boldsymbol{A}$ are randomly generated as in the initialization step, and $\boldsymbol{D}_0'', \ldots, \boldsymbol{D}_L''$ are diagonal matrices such that $\|\boldsymbol{D}_l''\|_2 \le 1$ for all $l \in [L]$ and $\boldsymbol{D}_l''$ is deterministic given $\{\boldsymbol{W}_a^{(0)} : a \le l\}$, and $\boldsymbol{W}_1', \ldots, \boldsymbol{W}_L' \in \mathbb{R}^{m \times m}$ are perturbation matrices with $\|\boldsymbol{W}_l'\|_2 < \tau\omega$ for all $l \in [L-1]$ for some $\omega < 1$. Then with probability at least $1 - (L) \cdot \exp(-\Omega(m))$ over the initialization randomness we have*

$$\|(\boldsymbol{I} + \tau \boldsymbol{W}_b^{(0)} + \tau \boldsymbol{W}_b')\boldsymbol{D}_{b-1}'' \cdots \boldsymbol{D}_a''(\boldsymbol{I} + \tau \boldsymbol{W}_a^{(0)} + \tau \boldsymbol{W}_a')\|_2 \le O(1). \tag{26}$$

*Proof.* This proof is based on the result of Theorem 1. From Theorem 1, we know for any $1 \le a \le b < L$

$$\|(\boldsymbol{I} + \tau \boldsymbol{W}_b^{(0)})\boldsymbol{D}_{b-1}'' \cdots \boldsymbol{D}_a''(\boldsymbol{I} + \tau \boldsymbol{W}_a^{(0)})\|_2 \le 1 + c.$$

Then we have

$$\|(\boldsymbol{I} + \tau \boldsymbol{W}_b^{(0)} + \tau \boldsymbol{W}_b')\boldsymbol{D}_{b-1}'' \cdots \boldsymbol{D}_a''(\boldsymbol{I} + \tau \boldsymbol{W}_a^{(0)} + \tau \boldsymbol{W}_a')\|_2$$
$$\le \sum_{j=0}^{b-a+1} \binom{b-a+1}{j} (\tau\|\boldsymbol{W}'\|)^j (1+c)^{j+1} \le (1+c) \cdot (1 + (1+c)\tau^2)^{b-a+1} \le O(1+c),$$

due to the assumption $\|\boldsymbol{W}_l'\| \le \tau\omega$ for $l \in [L-1]$ and $\omega < 1, \tau \le 1/\Omega(\sqrt{L})$. $\qquad\square$

We also have small changes on the output vector of each layer after perturbation.

**Lemma 4.** *Suppose for $\omega \le O(1)$, $\tau^2 L \le 1$, $\|\boldsymbol{W}_L'\|_2 \le \omega$ and $\|\boldsymbol{W}_l'\|_2 \le \tau\omega$ for $l \in [L-1]$. Then with probability at least $1 - \exp(-\Omega(m\omega^{2/3}))$, the following bounds on $h_{i,l}'$ and $\boldsymbol{D}_{i,l}'$ hold for all $i \in [n]$ and all $l \in [L-1]$,*

$$\|h_{i,l}'\| \le O(\tau^2 L\omega), \quad \|\boldsymbol{D}_{i,l}'\|_0 \le O\left(m(\omega\tau L)^{2/3}\right), \quad \|h_{i,L}'\| \le O(\omega), \quad \|\boldsymbol{D}_{i,L}'\|_0 \le O\left(m\omega^{2/3}\right).$$

*Proof.* Fixing $i$ and ignoring the subscript in $i$, by Claim 8.2 in Allen-Zhu et al. (2018b), for $l \in [L-1]$, there exists $\boldsymbol{D}_l''$ such that $|(\boldsymbol{D}_l'')_{k,k}| \le 1$ and

$$h_l' = \boldsymbol{D}_l'' \left((\boldsymbol{I} + \tau \boldsymbol{W}_l + \tau \boldsymbol{W}_l')h_{l-1} - (\boldsymbol{I} + \tau \boldsymbol{W}_l)h_{l-1}^{(0)}\right)$$
$$= \boldsymbol{D}_l'' \left((\boldsymbol{I} + \tau \boldsymbol{W}_l + \tau \boldsymbol{W}_l')h_{l-1}' + \tau \boldsymbol{W}_l' h_{l-1}^{(0)}\right)$$
$$= \boldsymbol{D}_l''(\boldsymbol{I} + \tau \boldsymbol{W}_l + \tau \boldsymbol{W}_l')\boldsymbol{D}_{l-1}''(\boldsymbol{I} + \tau \boldsymbol{W}_{l-1} + \tau \boldsymbol{W}_{l-1}')h_{l-2}'$$
$$\quad + \tau \boldsymbol{D}_l''(\boldsymbol{I} + \tau \boldsymbol{W}_l + \tau \boldsymbol{W}_l')\boldsymbol{D}_{l-1}'' \boldsymbol{W}_{l-1}' h_{l-2}^{(0)} + \tau \boldsymbol{D}_l'' \boldsymbol{W}_l' h_{l-1}^{(0)}$$
$$= \cdots$$
$$= \sum_{a=1}^{l} \tau \boldsymbol{D}_l''(\boldsymbol{I} + \tau \boldsymbol{W}_l + \tau \boldsymbol{W}_l') \cdots \boldsymbol{D}_{a+1}''(\boldsymbol{I} + \tau \boldsymbol{W}_{a+1} + \tau \boldsymbol{W}_{a+1}')\boldsymbol{D}_a'' \boldsymbol{W}_a' h_a^{(0)}. \tag{27}$$

We claim that

$$\|h_l'\| \le O(\tau^2 L\omega) \tag{28}$$

due to the fact $\|\boldsymbol{D}_l''\|_2 \le 1$ and the assumption $\|\boldsymbol{W}_l'\|_2 \le \tau\omega$ for $l \in [L-1]$. This implies that $\|h_{i,l}'\|, \|g_{i,l}'\| \le O(\tau^2 L\omega)$ for all $l \in [L-1]$ and for all $i$ with probability at least $1 - O(nL) \cdot \exp(-\Omega(m))$. One step further, we have $\|h_L'\|, \|g_L'\| \le O(\omega)$.

As for the sparsity $\|\boldsymbol{D}'_l\|_0$, we have $\|\boldsymbol{D}'_l\|_0 \leq O(m(\omega\tau L)^{2/3})$ for every $l = [L-1]$ and $\|\boldsymbol{D}'_L\|_0 \leq O(m\omega^{2/3})$.

The argument is as follows (adapt from the Claim 5.3 in Allen-Zhu et al. (2018b)).

We first study the case for $l \in [L-1]$. We observe that if $(\boldsymbol{D}'_l)_{j,j} \neq 0$ one must have

$$|(g'_l)_j| > |(g_l^{(0)})_j|.$$

We note that $(g_l^{(0)})_j = (h_{l-1}^{(0)} + \tau \boldsymbol{W}_l^{(0)} h_{l-1}^{(0)})_j \sim \mathcal{N}\left((h_{l-1}^{(0)})_j, \frac{2\tau^2 \|h_{l-1}^{(0)}\|^2}{m}\right)$. Let $\xi \leq \frac{1}{2\sqrt{m}}$ be a parameter to be chosen later. Let $S_1 \subseteq [m]$ be a index set satisfying $S_1 := \{j : |(g_l^{(0)})_j| \leq \xi\tau\}$. We have $\mathbb{P}\{|(g_l^{(0)})_j| \leq \xi\tau\} \leq O(\xi\sqrt{m})$ for each $j \in [m]$. By Chernoff bound, with probability at least $1 - \exp(-\Omega(m^{3/2}\xi))$ we have

$$|S_1| \leq O(\xi m^{3/2}).$$

Let $S_2 := \{j : j \notin S_1, \text{ and } (\boldsymbol{D}'_l)_{j,j} \neq 0\}$. Then for $j \in S_2$, we have $|(g'_l)_j| > \xi\tau$. As we have proved that $\|g'_l\| \leq O(\tau^2 L\omega)$, we have

$$|S_2| \leq \frac{\|g'_l\|^2}{(\xi\tau)^2} = O((\omega\tau L)^2/\xi^2).$$

Choosing $\xi$ to minimize $|S_1| + |S_2|$, we have $\xi = (\omega\tau L)^{2/3}/\sqrt{m}$ and consequently, $\|\boldsymbol{D}'_l\|_0 \leq O(m(\omega\tau L)^{2/3})$. Similarly, we have $\|\boldsymbol{D}'_L\|_0 \leq O(m\omega^{2/3})$. $\square$

We next prove that the norm of a sparse vector after the ResNet mapping.

**Lemma 5.** *If $s \geq \Omega(d/\log m)$ and $\tau \leq 1/\Omega(\sqrt{L})$, then for all $i \in [n]$ and $a \in [L]$ and for all $s$-sparse vectors $u \in \mathbb{R}^m$ and for all $v \in \mathbb{R}^d$, the following bound holds with probability at least $1 - (nL) \cdot \exp(-\Omega(s\log m))$*

$$|v^T \boldsymbol{B} \boldsymbol{D}_{i,L} \boldsymbol{W}_L \boldsymbol{D}_{i,L-1}(\boldsymbol{I} + \tau\boldsymbol{W}_{L-1}) \cdots \boldsymbol{D}_{i,a}(\boldsymbol{I} + \tau\boldsymbol{W}_a)u| \leq O\left(\frac{\sqrt{s\log m}}{\sqrt{d}}\|u\|\|v\|\right), \quad (29)$$

*where $\boldsymbol{D}_{i,a}$ is diagonal matrix with value 0 or 1 and it is independent of $\boldsymbol{W}_b$ for any $b \in (a, L]$.*

*Proof.* For any fixed vector $u \in \mathbb{R}^m$, $\|\boldsymbol{D}_{i,L}\boldsymbol{W}_L\boldsymbol{D}_{i,L-1}(\boldsymbol{I} + \tau\boldsymbol{W}_{L-1}) \cdots \boldsymbol{D}_{i,a}(\boldsymbol{I} + \tau\boldsymbol{W}_a)u\| \leq 1.1\|u\|$ holds with probability at least $1 - \exp(-\Omega(m))$ (over the randomness of $\boldsymbol{W}_l, l \in [L]$).

On the above event, for a fixed vector $v \in \mathbb{R}^d$ and any fixed $\boldsymbol{W}_l$ for $l \in [L]$, the randomness only comes from $\boldsymbol{B}$, then $v^T \boldsymbol{B} \boldsymbol{D}_{i,L} \boldsymbol{W}_L \boldsymbol{D}_{i,L-1}(\boldsymbol{I} + \tau\boldsymbol{W}_{L-1}) \cdots \boldsymbol{D}_{i,a}(\boldsymbol{I} + \tau\boldsymbol{W}_a)u$ is a Gaussian variable with mean 0 and variance no larger than $O(\|u\| \cdot \|v\|/\sqrt{d})$. Hence

$$\mathbb{P}\{|v^T \boldsymbol{B} \boldsymbol{D}_{i,L} \boldsymbol{W}_L \boldsymbol{D}_{i,L-1}(\boldsymbol{I} + \tau\boldsymbol{W}_{L-1}) \cdots \boldsymbol{D}_{i,a}(\boldsymbol{I} + \tau\boldsymbol{W}_a)u| \geq \sqrt{s\log m} \cdot \Omega(\|u\|\|v\|/\sqrt{d})\}$$
$$= \text{erfc}(\Omega(\sqrt{s\log m})) \leq \exp(-\Omega(s\log m)).$$

Take $\epsilon$-net over all $s$-sparse vectors of $u$ and all $d$-dimensional vectors of $v$, if $s \geq \Omega(d/\log m)$ then with probability $1 - \exp(-\Omega(s\log m))$ the claim holds for all $s$-sparse vectors of $u$ and all $d$-dimensional vectors of $v$. Further taking the union bound over all $i \in [n]$ and $a \in [L]$, the lemma is proved. $\square$

## D  GRADIENT LOWER/UPPER BOUNDS AND THEIR PROOFS

Because the gradient is pathological and data-dependent, in order to build bound on the gradient, we need to consider all possible point and all cases of data. Hence we first introduce an arbitrary loss vector and then the gradient bound can be obtained by taking a union bound.

We define the $\mathsf{BP}_{\overrightarrow{\boldsymbol{W}},i}(v,\cdot)$ operator. It back-propagates a vector $v$ to the $\cdot$ which could be the intermediate output $h_l$ or the parameter $\boldsymbol{W}_l$ at the specific layer $l$ using the forward propagation state of input $i$ through the network with parameter $\overrightarrow{\boldsymbol{W}}$. Specifically,

$$\mathsf{BP}_{\overrightarrow{\boldsymbol{W}},i}(v,h_l) := (\boldsymbol{I}+\tau\boldsymbol{W}_{l+1})^T\boldsymbol{D}_{i,l+1}\cdots(\boldsymbol{I}+\tau\boldsymbol{W}_{L-1})^T\boldsymbol{D}_{i,L-1}\boldsymbol{W}_L^T\boldsymbol{D}_{i,L}\boldsymbol{B}^Tv,$$

$$\mathsf{BP}_{\overrightarrow{\boldsymbol{W}},i}(v,\boldsymbol{W}_l) := \tau\left(\boldsymbol{D}_{i,l}(\boldsymbol{I}+\tau\boldsymbol{W}_{l+1})^T\cdots(\boldsymbol{I}+\tau\boldsymbol{W}_{L-1})^T\boldsymbol{D}_{i,L-1}\boldsymbol{W}_L^T\boldsymbol{D}_{i,L}\boldsymbol{B}^Tv\right)h_{i,l-1}^T \quad \forall l\in[L-1],$$

$$\mathsf{BP}_{\overrightarrow{\boldsymbol{W}},i}(v,\boldsymbol{W}_L) := \left(\boldsymbol{D}_{i,L}\boldsymbol{B}^Tv\right)h_{i,L-1}^T.$$

Moreover, we introduce

$$\mathsf{BP}_{\overrightarrow{\boldsymbol{W}}}(\overrightarrow{v},\boldsymbol{W}_l) := \sum_{i=1}^{n}\mathsf{BP}_{\overrightarrow{\boldsymbol{W}},i}(v_i,\boldsymbol{W}_l) \quad \forall l\in[L],$$

where $\overrightarrow{v}$ is composed of $n$ vectors $v_i$ for $i\in[n]$. If $v_i$ is the error signal of input $i$, then $\nabla_{\boldsymbol{W}_l}F_i(\overrightarrow{\boldsymbol{W}}) = \mathsf{BP}_{\overrightarrow{\boldsymbol{W}},i}(\boldsymbol{B}h_{i,L}-y_i^*,\boldsymbol{W}_l)$.

## D.1 GRADIENT UPPER BOUND

**Theorem 3.** *With probability at least $1-(nL)\cdot\exp(-\Omega(m))$ over the randomness of $\overrightarrow{\boldsymbol{W}}^{(0)},\boldsymbol{A},\boldsymbol{B}$, it satisfies for every $l\in[L-1]$, every $i\in[n]$, and every $\overrightarrow{\boldsymbol{W}}$ with $\|\overrightarrow{\boldsymbol{W}}-\overrightarrow{\boldsymbol{W}}^{(0)}\|_2 \le \omega$ for $\omega\in[0,1]$,*

$$\|\nabla_{\boldsymbol{W}_l}F_i(\overrightarrow{\boldsymbol{W}})\|_F^2 \le O\left(\frac{F_i(\overrightarrow{\boldsymbol{W}})}{d}\times\tau^2 m\right), \qquad \|\nabla_{\boldsymbol{W}_L}F_i(\overrightarrow{\boldsymbol{W}})\|_F^2 \le O\left(\frac{F_i(\overrightarrow{\boldsymbol{W}})}{d}\times m\right). \quad (4)$$

*Proof.* For each $i\in[n]$, we have

$$\left\|\mathsf{BP}_{\overrightarrow{\boldsymbol{W}}}(v_i,\boldsymbol{W}_L)\right\|_F = \left\|\boldsymbol{D}_{i,L}\left(\boldsymbol{B}^Tv_i\right)h_{i,L-1}^T\right\|_F = \left\|\boldsymbol{D}_{i,L}\left(\boldsymbol{B}^Tv_i\right)\right\|\left\|h_{i,L-1}^T\right\| \le O(\sqrt{m/d})\|v_i\|.$$

Similarly, we have for $l\in[L-1]$,

$$\left\|\mathsf{BP}_{\overrightarrow{\boldsymbol{W}}}(v_i,\boldsymbol{W}_l)\right\|_F = \tau\left(\boldsymbol{D}_{i,l}(\boldsymbol{I}+\tau\boldsymbol{W}_{l+1})^T\cdots(\boldsymbol{I}+\tau\boldsymbol{W}_{L-1})^T\boldsymbol{D}_{i,L-1}\boldsymbol{W}_L^T\boldsymbol{D}_{i,L}\boldsymbol{B}^Tv_i\right)h_{i,l-1}^T$$
$$\le O(\tau\sqrt{m/d})\|v_i\|.$$

The above upper bounds hold for the initialization $\overrightarrow{\boldsymbol{W}}^{(0)}$ because of Lemma 1 and Lemma 2. They also hold for all the $\overrightarrow{\boldsymbol{W}}$ such that $\|\overrightarrow{\boldsymbol{W}}-\overrightarrow{\boldsymbol{W}}^{(0)}\|_2 \le \omega$ due to Lemma 4 and Lemma 3.

Finally, taking $\epsilon-$net over all possible vectors $\overrightarrow{v}=(v_1,\ldots,v_n)\in(\mathbb{R}^d)^n$, we prove that the above bounds holds for all $\overrightarrow{v}$. In particular, we can now plug in the choice of $v_i=\boldsymbol{B}h_{i,L}-y_i^*$ and obtain the desired bounds on the true gradients. $\qquad\square$

## D.2 GRADIENT LOWER BOUND

**Theorem 7.** *Let $\omega = O\left(\frac{\delta^{3/2}}{n^3\log^3 m}\right)$. With probability at least $1-\exp(-\Omega(m\omega^{2/3}))$ over the randomness of $\overrightarrow{\boldsymbol{W}}^{(0)},\boldsymbol{A},\boldsymbol{B}$, it satisfies for every $\overrightarrow{\boldsymbol{W}}$ with $\|\overrightarrow{\boldsymbol{W}}-\overrightarrow{\boldsymbol{W}}^{(0)}\|_2 \le \omega$,*

$$\|\nabla_{\boldsymbol{W}_L}F(\overrightarrow{\boldsymbol{W}})\|_F^2 \ge \Omega\left(\frac{F(\overrightarrow{\boldsymbol{W}})}{dn/\delta}\times m\right). \quad (30)$$

This gradient lower bound on $\|\nabla_{\boldsymbol{W}_L}F(\overrightarrow{\boldsymbol{W}})\|_F^2$ acts like the gradient dominance condition (Zou and Gu, 2019; Allen-Zhu et al., 2018b) except that our range on $\omega$ does not depend on the depth $L$.

*Proof.* The gradient lower-bound at the initialization is given in (Allen-Zhu et al., 2018b, Section 6.2) and (Zou and Gu, 2019, Lemma 4.1) via the smoothed analysis (Spielman and Teng, 2004): with

high probability the gradient is lower-bounded, although the worst case it might be 0. We adopt the same proof for (Zou and Gu, 2019, Lemma 4.1) based on two preconditioned results Theorem 2 and Lemma 6. We shall not repeat it here.

Now suppose that we have $\|\nabla_{\boldsymbol{W}_L} F(\overrightarrow{\boldsymbol{W}}^{(0)})\|_F^2 \geq \Omega\left(\frac{F(\overrightarrow{\boldsymbol{W}}^{(0)})}{dn/\delta} \times m\right)$. We next bound the change of the gradient after perturbing the parameter. Recall that

$$\mathsf{BP}_{\overrightarrow{\boldsymbol{W}}^{(0)}}(\overrightarrow{v}, \boldsymbol{W}_L) - \mathsf{BP}_{\overrightarrow{\boldsymbol{W}}}(\overrightarrow{v}, \boldsymbol{W}_L) = \sum_{i=1}^{n}\left((v_i^T \boldsymbol{B}\boldsymbol{D}_{i,L}^{(0)})^T (h_{i,L-1}^{(0)})^T - (v_i^T \boldsymbol{B}\boldsymbol{D}_{i,L})^T (h_{i,L-1})^T\right)$$

By Lemma 4 and Lemma 5, we know,

$$\|v_i^T \boldsymbol{B}\boldsymbol{D}_{i,L}^{(0)} - v_i^T \boldsymbol{B}\boldsymbol{D}_{i,L}\| \leq O(\sqrt{m\omega^{2/3}}/\sqrt{d}) \cdot \|v_i\|.$$

Furthermore, we know

$$\|v_i^T \boldsymbol{B}\boldsymbol{D}_{i,L}\| \leq O(\sqrt{m/d}) \cdot \|v_i\|.$$

By Theorem 2 and Lemma 4, we have

$$\|h_{i,L-1}^{(0)}\| \leq 1.01 \quad \text{and} \quad \|h_{i,L-1} - h_{i,L-1}^{(0)}\| \leq O(\omega).$$

Combing the above bounds together, we have

$$\|\mathsf{BP}_{\overrightarrow{\boldsymbol{W}}^{(0)}}(\overrightarrow{v}, \boldsymbol{W}_L) - \mathsf{BP}_{\overrightarrow{\boldsymbol{W}}}(\overrightarrow{v}, \boldsymbol{W}_L)\|_F^2 \leq n\|\overrightarrow{v}\|^2 \cdot O(\sqrt{m\omega^{2/3}/d} + \omega\sqrt{m/d})^2 \leq n\|\overrightarrow{v}\|^2 \cdot O\left(\frac{m}{d}\omega^{2/3}\right)$$

Hence the gradient lower bound still holds for $\overrightarrow{\boldsymbol{W}}$ given $\omega < O\left(\frac{\delta^{3/2}}{n^3}\right)$.

Finally, taking $\epsilon$−net over all possible vectors $\overrightarrow{v} = (v_1, \ldots, v_n) \in (\mathbb{R}^d)^n$, we prove that the above gradient lower bound holds for all $\overrightarrow{v}$. In particular, we can now plug in the choice of $v_i = \boldsymbol{B}h_{i,L} - y_i^*$ and it implies our desired bounds on the true gradients. $\square$

The gradient lower bound requires the following property.

**Lemma 6.** *For any $\delta$ and any pair $(x_i, x_j)$ satisfying $\|x_i - x_j\|_2 \geq \delta$, then $\|h_{i,l} - h_{j,l}\| \geq \Omega(\delta)$ holds for all $l \in [L]$ with probability at least $1 - O(n^2 L) \cdot \exp(-\Omega(\log^2 m))$ for $\tau \leq 1/\Omega(\sqrt{L}\log m)$ and $m \geq \Omega(\tau^2 L^2 \delta^{-2})$.*

The proof of Lemma 6 follows that of (Allen-Zhu et al., 2018b, Appendix C.1) given the condition that $m \geq \Omega(\tau^2 L^2 \delta^{-2})$.

# E   SEMI-SMOOTHNESS FOR $\tau \leq 1/\Omega(\sqrt{L})$

With the help of Theorem 3 and several improvements, we can obtain a tighter bound on the semi-smoothness condition of the objective function.

**Theorem 8.** *Let $\omega \in \left[\Omega\left(\left(\frac{d}{m\log m}\right)^{3/2}\right), O(1)\right]$ and $\overrightarrow{\boldsymbol{W}}^{(0)}, \boldsymbol{A}, \boldsymbol{B}$ be at random initialization and $\tau^2 L \leq 1$. With probability at least $1 - \exp(-\Omega(m\omega^{2/3}))$ over the randomness of $\overrightarrow{\boldsymbol{W}}^{(0)}, \boldsymbol{A}, \boldsymbol{B}$, we have for every $\overrightarrow{\breve{\boldsymbol{W}}} \in (\mathbb{R}^{m\times m})^L$ with $\|\breve{\boldsymbol{W}}_L - \boldsymbol{W}_L^{(0)}\|_2 \leq \omega$ and $\|\breve{\boldsymbol{W}}_l - \boldsymbol{W}_l^{(0)}\|_2 \leq \tau\omega$ for $l \in [L-1]$, and for every $\overrightarrow{\boldsymbol{W}}' \in (\mathbb{R}^{m\times m})^L$ with $\|\boldsymbol{W}'_L\|_2 \leq \omega$ and $\|\boldsymbol{W}'_l\|_2 \leq \tau\omega$ for $l \in [L-1]$, we have*

$$F(\overrightarrow{\breve{\boldsymbol{W}}} + \overrightarrow{\boldsymbol{W}}') \leq F(\overrightarrow{\breve{\boldsymbol{W}}}) + \langle \nabla F(\overrightarrow{\breve{\boldsymbol{W}}}), \overrightarrow{\boldsymbol{W}}'\rangle + O\left(\frac{nm}{d}\right)\|\overrightarrow{\boldsymbol{W}}'\|_F^2$$

$$+ O\left(\sqrt{\frac{mnL\omega^{2/3}}{d}} \cdot (\tau L)^{4/3}\right)\|\overrightarrow{\boldsymbol{W}}'\|_F \sqrt{F(\overrightarrow{\breve{\boldsymbol{W}}})}. \tag{31}$$

Before going to the proof of the theorem, we introduce a lemma.

**Lemma 7.** *There exist diagonal matrices $\boldsymbol{D}''_{i,l} \in \mathbb{R}^{m \times m}$ with entries in [-1,1] such that $\forall i \in [n], \forall l \in [L-1]$,*

$$h_{i,l} - \breve{h}_{i,l} = \sum_{a=1}^{l} (\breve{\boldsymbol{D}}_{i,l} + \boldsymbol{D}''_{i,l})(\boldsymbol{I} + \tau \breve{\boldsymbol{W}}_l) \cdots (\boldsymbol{I} + \tau \breve{\boldsymbol{W}}_{a+1})(\breve{\boldsymbol{D}}_{i,a} + \boldsymbol{D}''_{i,a}) \tau \boldsymbol{W}'_a h_{i,a-1}, \quad (32)$$

*and*

$$
\begin{aligned}
h_{i,L} - \breve{h}_{i,L} =& (\breve{\boldsymbol{D}}_{i,L} + \boldsymbol{D}''_{i,L}) \boldsymbol{W}'_L h_{i,L-1} \\
& + \sum_{a=1}^{L-1} (\breve{\boldsymbol{D}}_{i,L} + \boldsymbol{D}''_{i,L}) \breve{\boldsymbol{W}}_L \cdots (\boldsymbol{I} + \tau \breve{\boldsymbol{W}}_{a+1})(\breve{\boldsymbol{D}}_{i,a} + \boldsymbol{D}''_{i,a}) \tau \boldsymbol{W}'_a h_{i,a-1}. \quad (33)
\end{aligned}
$$

*Furthermore, we then have $\forall l \in [L-1], \|h_{i,l} - \breve{h}_{i,l}\| \leq O(\tau^2 L \omega), \|\boldsymbol{D}''_{i,l}\|_0 \leq O(m(\omega \tau L)^{2/3})$, and $\|h_{i,L} - \breve{h}_{i,L}\| \leq O((1 + \tau\sqrt{L})\|\boldsymbol{W}'\|_F), \|\boldsymbol{D}''_{i,L}\|_0 \leq O(m\omega^{2/3})$ and*

$$\|\boldsymbol{B}h_{i,L} - \boldsymbol{B}\breve{h}_{i,L}\| \leq O(\sqrt{m/d})\|\boldsymbol{W}'\|_F$$

*hold with probability $1 - \exp(-\Omega(m\omega^{2/3}))$ given $\|\boldsymbol{W}'_L\|_2 \leq \omega, \|\boldsymbol{W}'_l\|_2 \leq \tau\omega$ for $l \in [L-1]$ and $\omega \leq O(1), \tau\sqrt{L} \leq 1$.*

*Proof.* The proof can adapt from the proof of Claim 8.2 in Allen-Zhu et al. (2018b) and the proof of Lemma 4. ☐

*Proof of Theorem 8.* First of all, we know that $\breve{loss}_i := \boldsymbol{B}\breve{h}_{i,L} - y_i^*$

$$
\begin{aligned}
\frac{1}{2}\|\boldsymbol{B}h_{i,L} - y_i^*\|^2 &= \frac{1}{2}\|\breve{loss}_i + \boldsymbol{B}(h_{i,L} - \breve{h}_{i,L})\|^2 \\
&= \frac{1}{2}\|\breve{loss}_i\|^2 + \breve{loss}_i^T \boldsymbol{B}(h_{i,L} - \breve{h}_{i,L}) + \frac{1}{2}\|\boldsymbol{B}(h_{i,L} - \breve{h}_{i,L})\|^2, \quad (34)
\end{aligned}
$$

and

$$\nabla_{\boldsymbol{W}_l} F(\overrightarrow{\boldsymbol{W}}) = \sum_{i=1}^{n} (loss_i^T \boldsymbol{B} \boldsymbol{D}_{i,L} \boldsymbol{W}_L \cdots \boldsymbol{D}_{i,l+1}(\boldsymbol{I} + \tau \boldsymbol{W}_l) \boldsymbol{D}_{i,l})^T (\tau h_{i,l-1})^T. \quad (35)$$

$$\nabla_{\boldsymbol{W}_L} F(\overrightarrow{\boldsymbol{W}}) = \sum_{i=1}^{n} (loss_i^T \boldsymbol{B} \boldsymbol{D}_{i,L})^T (h_{i,l-1})^T. \quad (36)$$

Then,

$$
\begin{aligned}
& F(\overrightarrow{\breve{\boldsymbol{W}}} + \overrightarrow{\boldsymbol{W}}') - F(\overrightarrow{\breve{\boldsymbol{W}}}) - \langle \nabla F(\overrightarrow{\breve{\boldsymbol{W}}}), \overrightarrow{\boldsymbol{W}}' \rangle \\
&= -\langle \nabla F(\overrightarrow{\breve{\boldsymbol{W}}}), \overrightarrow{\boldsymbol{W}}' \rangle + \frac{1}{2} \sum_{i=1}^{n} \|\boldsymbol{B}h_{i,L} - y_i^*\|^2 - \|\boldsymbol{B}\breve{h}_{i,L} - y_i^*\|^2 \\
&= -\sum_{l=1}^{L} \langle \nabla_{\boldsymbol{W}_l} F(\overrightarrow{\breve{\boldsymbol{W}}}), \boldsymbol{W}'_l \rangle + \sum_{i=1}^{n} \breve{loss}_i^T \boldsymbol{B}(h_{i,L} - \breve{h}_{i,L}) + \frac{1}{2}\|\boldsymbol{B}(h_{i,L} - \breve{h}_{i,L})\|^2 \\
&\overset{(a)}{=} \frac{1}{2} \sum_{i=1}^{n} \|\boldsymbol{B}(h_{i,L} - \breve{h}_{i,L})\|^2 + \sum_{i=1}^{n} \breve{loss}_i^T \boldsymbol{B}\left((\breve{\boldsymbol{D}}_{i,L} + \boldsymbol{D}''_{i,L}) \boldsymbol{W}'_L h_{i,L-1} - (\breve{\boldsymbol{D}}_{i,L}) \boldsymbol{W}'_L \breve{h}_{i,L-1}\right) \\
&\quad + \sum_{i=1}^{n} \sum_{l=1}^{L-1} \breve{loss}_i^T \boldsymbol{B}\Big((\breve{\boldsymbol{D}}_{i,L} + \boldsymbol{D}''_{i,L}) \breve{\boldsymbol{W}}_L \cdots (\boldsymbol{I} + \tau \breve{\boldsymbol{W}}_{l+1})(\breve{\boldsymbol{D}}_{i,l} + \boldsymbol{D}''_{i,l}) \tau \boldsymbol{W}'_l h_{i,l-1} \\
&\quad - \breve{\boldsymbol{D}}_{i,L} \breve{\boldsymbol{W}}_L \cdots (\boldsymbol{I} + \tau \breve{\boldsymbol{W}}_{l+1}) \breve{\boldsymbol{D}}_{i,l} \boldsymbol{W}'_l (\tau \breve{h}_{i,l-1})\Big), \quad (37)
\end{aligned}
$$

where (a) is due to Lemma 7.

We next bound the RHS of equation 37. We first use Lemma 7 to get

$$\|\boldsymbol{B}(h_{i,L} - \breve{h}_{i,L})\| \le O(\sqrt{m/d})\|\boldsymbol{W}'\|_F. \tag{38}$$

Next we calculate that for $l = L$,

$$\left| \breve{loss}_i^T \boldsymbol{B} \left( (\breve{\boldsymbol{D}}_{i,L} + \boldsymbol{D}''_{i,L})\boldsymbol{W}'_L h_{i,L-1} - (\breve{\boldsymbol{D}}_{i,L})\boldsymbol{W}'_L \breve{h}_{i,L-1} \right) \right|$$

$$\le \left| \breve{loss}_i^T \boldsymbol{B} \left( \boldsymbol{D}''_{i,L} \boldsymbol{W}'_L h_{i,L-1} \right) \right| + \left| \breve{loss}_i^T \boldsymbol{B} \left( \breve{\boldsymbol{D}}_{i,L} \boldsymbol{W}'_L (h_{i,L-1} - \breve{h}_{i,L-1}) \right) \right|. \tag{39}$$

For the first term, by Lemma 5 and Lemma 7, we have

$$\left| \breve{loss}_i^T \boldsymbol{B} \left( \boldsymbol{D}''_{i,L} \boldsymbol{W}'_L h_{i,L-1} \right) \right| \le O\left( \frac{\sqrt{m}\omega^{2/3}}{\sqrt{d}} \right) \|\breve{loss}_i\| \cdot \|\boldsymbol{W}'_L h_{i,L-1}\|$$

$$\le O\left( \frac{\sqrt{m}\omega^{2/3}}{\sqrt{d}} \right) \|\breve{loss}_i\| \cdot \|\boldsymbol{W}'_L\|_2, \tag{40}$$

where the last inequality is due to $\|h_{i,L-1}\| \le O(1)$. For the second term, by Lemma 7 we have

$$\left| \breve{loss}_i^T \boldsymbol{B} \left( \breve{\boldsymbol{D}}_{i,L} \boldsymbol{W}'_L (h_{i,L-1} - \breve{h}_{i,L-1}) \right) \right|$$

$$\le \|\breve{loss}_i\| \cdot \left\| \boldsymbol{B}\breve{\boldsymbol{D}}_{i,L} \right\|_2 \cdot \|\boldsymbol{W}'_L\|_2 \|h_{i,L-1} - \breve{h}_{i,L-1}\|$$

$$\le \|\breve{loss}_i\| \cdot O\left( \frac{\omega\sqrt{m}}{\sqrt{d}} \right) \cdot \|\boldsymbol{W}'_L\|_2, \tag{41}$$

where the last inequality is due to the assumption $\|\boldsymbol{W}'_L\|_2 \le \omega$. Similarly for $l \in [L-1]$, we ignore the index $i$ for simplicity.

$$\left| \sum_{l=1}^{L-1} \breve{loss}^T \left( \boldsymbol{B}(\breve{\boldsymbol{D}}_L + \boldsymbol{D}''_L)\breve{\boldsymbol{W}}_L \cdots (\boldsymbol{I} + \tau\breve{\boldsymbol{W}}_{l+1})(\breve{\boldsymbol{D}}_l + \boldsymbol{D}''_l) - \boldsymbol{B}\breve{\boldsymbol{D}}_L\breve{\boldsymbol{W}}_L \cdots (\boldsymbol{I} + \tau\breve{\boldsymbol{W}}_{l+1})\breve{\boldsymbol{D}}_l \right) \boldsymbol{W}'_l(\tau h_{l-1}) \right|$$

$$= \left| \sum_{l=1}^{L-1} \breve{loss}^T \boldsymbol{B}\boldsymbol{D}''_L\breve{\boldsymbol{W}}_L(\boldsymbol{D}_{L-1} + \boldsymbol{D}''_{L-1})(\boldsymbol{I} + \tau\breve{\boldsymbol{W}}_{L-1}) \cdots (\boldsymbol{D}_l + \boldsymbol{D}''_l)(\tau\boldsymbol{W}'_l h_{l-1}) \right|$$

$$+ \left| \sum_{l=1}^{L-1} \sum_{a=l}^{L-1} \breve{loss}^T \boldsymbol{B}\breve{\boldsymbol{D}}_L\breve{\boldsymbol{W}}_L \cdots (\boldsymbol{I} + \tau\breve{\boldsymbol{W}}_{a+1})\boldsymbol{D}''_a(\boldsymbol{I} + \tau\breve{\boldsymbol{W}}_a) \cdots (\boldsymbol{D}_l + \boldsymbol{D}''_l)(\tau\boldsymbol{W}'_l h_{l-1}) \right|$$

$$+ \left| \sum_{l=1}^{L-1} \breve{loss}^T \boldsymbol{B}\breve{\boldsymbol{D}}_L\breve{\boldsymbol{W}}_L \cdots (\boldsymbol{I} + \tau\breve{\boldsymbol{W}}_{l+1})\breve{\boldsymbol{D}}_l\boldsymbol{W}'_l\tau(h_{l-1} - \breve{h}_{l-1}) \right| \tag{42}$$

We next bound the terms in equation 42 one by one. For the first term, by Lemma 5 and Lemma 7, we have

$$\left| \sum_{l=1}^{L-1} \breve{loss}^T \boldsymbol{B}\boldsymbol{D}''_L\breve{\boldsymbol{W}}_L(\boldsymbol{D}_{L-1} + \boldsymbol{D}''_{L-1})(\boldsymbol{I} + \tau\breve{\boldsymbol{W}}_{L-1}) \cdots (\boldsymbol{D}_l + \boldsymbol{D}''_l)(\tau\boldsymbol{W}'_l h_{l-1}) \right|$$

$$\le O\left( \frac{\sqrt{m}\omega^{2/3}}{\sqrt{d}} \right) \left\| \breve{loss} \right\| \cdot \left\| \sum_{l=1}^{L-1} \breve{\boldsymbol{W}}_L(\boldsymbol{D}_{L-1} + \boldsymbol{D}''_{L-1})(\boldsymbol{I} + \tau\breve{\boldsymbol{W}}_{L-1}) \cdots (\boldsymbol{D}_l + \boldsymbol{D}''_l)(\tau\boldsymbol{W}'_l h_{l-1}) \right\|$$

$$\overset{(a)}{\le} O\left( \frac{\sqrt{m}\omega^{2/3}}{\sqrt{d}} \right) \cdot \|\breve{loss}\| \cdot \tau\sqrt{L}\|\boldsymbol{W}'_{L-1:1}\|_F, \tag{43}$$

where (a) is due to the similar argument in the proof Lemma 7 and the fact $\left\| \breve{\boldsymbol{W}}_L(\boldsymbol{D}_{L-1} + \boldsymbol{D}''_{L-1})(\boldsymbol{I} + \tau\breve{\boldsymbol{W}}_{L-1}) \cdots (\boldsymbol{D}_l + \boldsymbol{D}''_l) \right\| = O(1)$ and $\|h_{l-1}\| = O(1)$ holds with high probability.

We have similar bound for the second term of equation 42

$$
\left| \sum_{l=1}^{L-1} \sum_{a=l}^{L-1} \breve{loss}^T \boldsymbol{B} \breve{\boldsymbol{D}}_L \breve{\boldsymbol{W}}_L \cdots (\boldsymbol{I} + \tau \breve{\boldsymbol{W}}_{a+1}) \boldsymbol{D}_a'' (\boldsymbol{I} + \tau \breve{\boldsymbol{W}}_a) \cdots (\boldsymbol{D}_l + \boldsymbol{D}_l'') (\tau \boldsymbol{W}_l' h_{l-1}) \right|
$$

$$
\leq O \left( \frac{\sqrt{m(\omega \tau L)^{2/3}}}{\sqrt{d}} \right) \cdot \|\breve{loss}\| \cdot \tau \sum_{a=1}^{L-1} \sqrt{a} \|\boldsymbol{W}_{a:1}\|_F
$$

$$
\leq O \left( \frac{\sqrt{m(\omega \tau L)^{2/3}}}{\sqrt{d}} \right) \cdot \|\breve{loss}\| \cdot \tau L^{3/2} \|\boldsymbol{W}_{L-1:1}\|_F. \tag{44}
$$

For the last term in equation 42, we have

$$
\left| \sum_{l=1}^{L-1} \breve{loss}^T \boldsymbol{B} \breve{\boldsymbol{D}}_L \breve{\boldsymbol{W}}_L \cdots (\boldsymbol{I} + \tau \breve{\boldsymbol{W}}_{l+1}) \breve{\boldsymbol{D}}_l \boldsymbol{W}_l' \tau (h_{l-1} - \breve{h}_{l-1}) \right|
$$

$$
\leq \|\breve{loss}\| \cdot O \left( \sqrt{m/d} \right) \cdot \sum_{l=1}^{L-1} \|\boldsymbol{W}_l'\|_2 \cdot \tau^3 L \omega
$$

$$
\leq \|\breve{loss}\| \cdot O \left( \sqrt{m/d} \right) \cdot \|\boldsymbol{W}_{L-1:1}'\|_F \cdot (\tau^2 L)^{3/2}, \tag{45}
$$

where is the last inequality is due to the bound on $\|h_{l-1} - \breve{h}_{l-1}\|_2$ in Lemma 7. Hence

$$
equation~42 \leq O \left( \frac{\sqrt{m(\omega \tau L)^{2/3}}}{\sqrt{d}} \right) \cdot \|\breve{loss}\| \cdot \tau L^{3/2} \|\boldsymbol{W}_{L-1:1}\|_F
$$

$$
\leq O \left( (\tau L)^{4/3} \frac{\sqrt{m L \omega^{2/3}}}{\sqrt{d}} \right) \cdot \|\breve{loss}\| \cdot \|\boldsymbol{W}_{L-1:1}'\|_F. \tag{46}
$$

Having all the above together and using triangle inequality, we have the result. $\square$

**Proposition 1** (Proposition 8.3 in in Allen-Zhu et al. (2018b)). *Given vectors $a, b \in \mathbb{R}^m$ and $\boldsymbol{D} \in \mathbb{R}^{m \times m}$ the diagonal matrix where $\boldsymbol{D}_{k,k} = \mathbf{1}_{a_k \geq 0}$. Then, there exists a diagonal matrix $\boldsymbol{D}'' \in \mathbb{R}^{m \times m}$ with*

- *$|\boldsymbol{D}_{k,k} + \boldsymbol{D}_{k,k}''| \leq 1$ and $|\boldsymbol{D}_{k,k}''| \leq 1$ for every $k \in [m]$,*

- *$\boldsymbol{D}_{k,k}'' \neq 0$ only when $\mathbf{1}_{a_k \geq 0} \neq \mathbf{1}_{b_k \geq 0}$,*

- *$\phi(a) - \phi(b) = (\boldsymbol{D} + \boldsymbol{D}'')(a - b)$.*

*Proof of Lemma 7.* Fixing index $i$ and ignoring the subscript in $i$ for simplicity, by Proposition 1, for each $l \in [L-1]$ there exists a $\boldsymbol{D}_l''$ such that $|(\boldsymbol{D}_l'')_{k,k}| \leq 1$ and

$$
h_l - \breve{h}_l = \phi((\boldsymbol{I} + \tau \breve{\boldsymbol{W}}_l + \tau \boldsymbol{W}_l') h_{l-1}) - \phi((\boldsymbol{I} + \tau \breve{\boldsymbol{W}}_l) \breve{h}_{l-1})
$$

$$
= (\breve{\boldsymbol{D}}_l + \boldsymbol{D}_l'') \left( (\boldsymbol{I} + \tau \breve{\boldsymbol{W}}_l + \tau \boldsymbol{W}_l') h_{l-1} - (\boldsymbol{I} + \tau \breve{\boldsymbol{W}}_l) \breve{h}_{l-1} \right)
$$

$$
= (\breve{\boldsymbol{D}}_l + \boldsymbol{D}_l'')(\boldsymbol{I} + \tau \breve{\boldsymbol{W}}_l)(h_{l-1} - \breve{h}_{l-1}) + (\breve{\boldsymbol{D}}_l + \boldsymbol{D}_l'') \tau \boldsymbol{W}_l' h_{l-1}
$$

$$
= \sum_{a=1}^{l} (\breve{\boldsymbol{D}}_l + \boldsymbol{D}_l'')(\boldsymbol{I} + \tau \breve{\boldsymbol{W}}_l) \cdots (\boldsymbol{I} + \tau \breve{\boldsymbol{W}}_{a+1})(\breve{\boldsymbol{D}}_a + \boldsymbol{D}_a'') \tau \boldsymbol{W}_a' h_{a-1}
$$

Then we have following properties. For $l \in [L-1]$, $\|h_l - \breve{h}_l\| \leq O(\tau^2 L \omega)$. This is because $\|(\breve{\boldsymbol{D}}_l + \boldsymbol{D}_l'')(\boldsymbol{I} + \tau \breve{\boldsymbol{W}}_l) \cdots (\boldsymbol{I} + \tau \breve{\boldsymbol{W}}_{a+1})(\breve{\boldsymbol{D}}_a + \boldsymbol{D}_a'')\| \leq 1.1$ from Lemma 3; $\|h_{a-1}\| \leq O(1)$ from Theorem 2; and the assumption $\|\boldsymbol{W}_l'\|_2 \leq \tau \omega$ for $l \in [L-1]$.

To have a tighter bound on $\|h_L - \breve{h}_L\|$, let us introduce $\boldsymbol{W}_b'' := \sum_{a=b}^{l} (\breve{\boldsymbol{D}}_l + \boldsymbol{D}_l'')(\boldsymbol{I} + \tau\breve{\boldsymbol{W}}_l)\cdots(\boldsymbol{I} + \tau\breve{\boldsymbol{W}}_{a+1})(\breve{\boldsymbol{D}}_a + \boldsymbol{D}_a'')\boldsymbol{W}_a'$, for $b = 1, ..., l$. Then we have

$$h_L - \breve{h}_L = \left[\boldsymbol{W}_L'', \boldsymbol{W}_{L-1}'', ..., \boldsymbol{W}_1''\right] [h_{L-1}^T, \tau h_{L-2}^T, ..., \tau h_0^T]^T. \tag{47}$$

It is easy to get

$$\|[\tau h_{l-1}^T, \tau h_{l-2}^T, ..., \tau h_0^T]^T\|_2 = \sqrt{\tau^2 \sum_{a=0}^{l-1} \|h_a\|^2} \leq \tau\sqrt{L} \cdot O(1),$$

where the inequality is because of $\|h_{a-1}\| \leq O(1)$ from Theorem 2. Next, we have

$$\left\|\left[\boldsymbol{W}_l'', \boldsymbol{W}_{l-1}'', ..., \boldsymbol{W}_1''\right]\right\|_2 = \left\|\left[\boldsymbol{W}_l'', \boldsymbol{W}_{l-1}'', ..., \boldsymbol{W}_1''\right]^T\right\|_2$$

$$\leq \sqrt{\sum_{a=1}^{l} \|(\boldsymbol{W}_l'')^T\|_2^2} \leq 1.1\sqrt{\sum_{a=1}^{l} \|(\boldsymbol{W}_l')^T\|_2^2} \leq 1.1\|\boldsymbol{W}_{l:1}'\|_F, \tag{48}$$

where the second inequality is from the definition of spectral norm, the third inequality is because of $\|(\breve{\boldsymbol{D}}_l + \boldsymbol{D}_l'')(\boldsymbol{I} + \tau\breve{\boldsymbol{W}}_l)\cdots(\boldsymbol{I} + \tau\breve{\boldsymbol{W}}_{a+1})(\breve{\boldsymbol{D}}_a + \boldsymbol{D}_a'')\| \leq 1.1$ from Lemma 3.

Hence we have $\|h_L - \breve{h}_L\| \leq O\left((1 + \tau\sqrt{L})\|\boldsymbol{W}'\|_F\right) = O\left(\|\boldsymbol{W}'\|_F\right)$ because of the assumption $\tau\sqrt{L} \leq 1$.

For $l \in [L]$, $\|\boldsymbol{D}_l''\|_0 \leq O(m\omega^{2/3})$. This is because $(\boldsymbol{D}_l'')_{k,k}$ is non-zero only at coordinates $k$ where $(\breve{g}_l)_k$ and $(g_l)_k$ have opposite signs, where it holds either $(\boldsymbol{D}_l^{(0)})_{k,k} \neq (\breve{\boldsymbol{D}}_l)_{k,k}$ or $(\boldsymbol{D}_l^{(0)})_{k,k} \neq (\boldsymbol{D}_l)_{k,k}$. Therefore by Lemma 4, we have $\|\boldsymbol{D}_l''\|_0 \leq O(m(\omega\tau L)^{2/3})$ if $\|\boldsymbol{W}_l'\|_2 \leq \tau\omega$.

$\square$

## F    PROOF FOR THEOREM 5

**Theorem 5.** *Suppose that the ResNet is defined as in Section 2 with $\tau \leq 1/\Omega(\sqrt{L}\log m)$ and training data satisfy Assumption 1. If the network width $m \geq \Omega(n^8 L^7 \delta^{-4} d \log^2 m)$, then with probability at least $1 - \exp(-\Omega(\log^2 m))$, gradient descent with learning rate $\eta = \Theta(\frac{d}{nm})$ finds a point $F(\overrightarrow{\boldsymbol{W}}) \leq \varepsilon$ in $T = \Omega(n^2 \delta^{-1} \log\frac{n\log^2 m}{\varepsilon})$ iterations.*

### F.1    CONVERGENCE RESULT FOR GD

*Proof.* Using Theorem 2 we have $\|h_{i,L}^{(0)}\|_2 \leq 1.1$ and then using the randomness of $\boldsymbol{B}$, it is easy to show that $\|\boldsymbol{B}h_{i,L}^{(0)} - y_i^*\|^2 \leq O(\log^2 m)$ with probability at least $1 - \exp(-\Omega(\log^2 m))$, and therefore

$$F(\overrightarrow{\boldsymbol{W}}^{(0)}) \leq O(n\log^2 m). \tag{49}$$

Assume that for every $t = 0, 1, \ldots, T-1$, the following holds,

$$\|\boldsymbol{W}_L^{(t)} - \boldsymbol{W}_L^{(0)}\|_F \leq \omega \triangleq O\left(\frac{\delta^{3/2}}{n^3 L^{7/2}}\right) \tag{50}$$

$$\|\boldsymbol{W}_l^{(t)} - \boldsymbol{W}_l^{(0)}\|_F \leq \tau\omega. \tag{51}$$

We shall prove the convergence of GD under the assumption equation 50 holds, so that previous statements can be applied. At the end, we shall verify that equation 50 is indeed satisfied.

Letting $\nabla_t = \nabla F(\overrightarrow{\boldsymbol{W}}^{(t)})$, we calculate that

$$F(\overrightarrow{\boldsymbol{W}}^{(t+1)}) \leq F(\overrightarrow{\boldsymbol{W}}^{(t)}) - \eta\|\nabla_t\|_F^2 + O(\eta^2 nm/d)\|\nabla_t\|_F^2 +$$

$$\eta\sqrt{F(\overrightarrow{\boldsymbol{W}}^{(t)})} \cdot O\left(\sqrt{\frac{mnL\omega^{2/3}}{d}}(\tau L)^{4/3}\right) \cdot \|\nabla_t\|_F$$

$$\leq \left(1 - \Omega\left(\frac{\eta\delta m}{dn}\right)\right)F(\overrightarrow{\boldsymbol{W}}^{(t)}), \tag{52}$$

where the first inequality uses Theorem 4, the second inequality uses the gradient upper bound in Theorem 3 and the last inequality uses the gradient lower bound in Theorem 7 and the choice of $\eta = O(d/(mn))$ and the assumption on $\omega$ equation 50. That is, after $T = \Omega(\frac{dn}{\eta \delta m}) \log \frac{n \log^2 m}{\epsilon}$ iterations $F(\overrightarrow{\boldsymbol{W}}^{(T)}) \le \epsilon$.

We need to verify for each $t$, equation 50 holds. Here we use a result from (Zou and Gu, 2019, Lemma 4.2) that states $\|\boldsymbol{W}_L^{(t)} - \boldsymbol{W}_L^{(0)}\|_2 \le O(\sqrt{\frac{n^2 d \log m}{m \delta}})$.

To guarantee the iterates fall into the region given by $\omega$ equation 50, we obtain a bound $m \ge n^8 \delta^{-4} dL^7 \log^2 m$.

$\square$

## G  TIGHTNESS OF $\tau = 1/\sqrt{L}$ AND THE PROOF OF THEOREM 4

**Theorem 4.** *For the ResNet defined and initialized as in Section 2, if $\tau \ge L^{-\frac{1}{2}+c}$, then in expectation*

$$\boldsymbol{E}\|h_L\|^2 > L^{2c}. \tag{5}$$

*Proof.* By induction we can show for any $k \in [m]$ and $l \in [L-1]$,

$$(h_l)_k \ge \phi\left(\sum_{a=1}^l (\tau \boldsymbol{W}_a h_{a-1})_k\right). \tag{53}$$

It is easy to verify $(h_1)_k = \phi\left((h_0)_k + (\tau \boldsymbol{W}_1 h_0)_k\right) \ge \phi\left((\tau \boldsymbol{W}_1 h_0)_k\right)$ because of $(h_0)_k \ge 0$.

Then assume $(h_l)_k \ge \phi\left(\sum_{a=1}^l (\tau \boldsymbol{W}_a h_{a-1})_k\right)$, we show it holds for $l+1$.

$$(h_{l+1})_k = \phi\left((h_l)_k + (\tau \boldsymbol{W}_{l+1} h_l)_k\right) \ge \phi\left(\phi\left(\sum_{a=1}^l (\tau \boldsymbol{W}_a h_{a-1})_k\right) + (\tau \boldsymbol{W}_{l+1} h_l)_k\right) \ge \phi\left(\sum_{a=1}^{l+1} (\tau \boldsymbol{W}_a h_{a-1})_k\right),$$

where the last inequality can be shown by case study.

Next we can compute the mean and variance of $\sum_{a=1}^l (\tau \boldsymbol{W}_a h_{a-1})_k$ by taking iterative conditioning. We have

$$\boldsymbol{E} \sum_{a=1}^l (\tau \boldsymbol{W}_a h_{a-1})_k = 0, \quad \boldsymbol{E}\left(\sum_{a=1}^l (\tau \boldsymbol{W}_a h_{a-1})_k\right)^2 = \frac{2\tau^2}{m} \sum_{a=1}^l \boldsymbol{E}\|h_{a-1}\|^2. \tag{54}$$

Moreover, $(\tau \boldsymbol{W}_a h_{a-1})_k$ are jointly Gaussian for all $a$ with mean 0 because $\boldsymbol{W}_a$'s are drawn from independent Gaussian distributions. We use $l = 2$ as an example to illustrate the conclusion, it can be generalized to other $l$. Assume that $h_0$ is fixed. First it is easy to verify that $(\tau \boldsymbol{W}_1 h_0)_k$ is Gaussian variable with mean 0 and $(\tau \boldsymbol{W}_2 h_1)_k | \boldsymbol{W}_1$ is also Gaussian variable with mean 0. Hence $[(\tau \boldsymbol{W}_1 h_0)_k, (\tau \boldsymbol{W}_2 h_1)_k]$ follows jointly Gaussian with mean vector $[0, 0]$. Thus $(\tau \boldsymbol{W}_1 h_0)_k + (\tau \boldsymbol{W}_2 h_1)_k$ is Gaussian with mean 0. By induction, we have $\sum_{a=1}^l (\tau \boldsymbol{W}_a h_{a-1})_k$ is Gaussian with mean 0. Then we have

$$\boldsymbol{E}\|h_l\|^2 \ge \sum_{k=1}^m \boldsymbol{E}\left(\phi\left(\sum_{a=1}^l (\tau \boldsymbol{W}_a h_{a-1})_k\right)\right)^2 = \sum_{k=1}^m \frac{1}{2} \boldsymbol{E}\left(\sum_{a=1}^l (\tau \boldsymbol{W}_a h_{a-1})_k\right)^2$$

$$= \sum_{k=1}^m \frac{\tau^2 \sum_{a=1}^l \mathbb{E}\left[\|h_{a-1}\|^2\right]}{m} = \tau^2 \sum_{a=1}^l \mathbb{E}\|h_{a-1}\|^2, \tag{55}$$

where the first step is due to equation 53, the second step is due to the symmetry of Gaussian distribution and the third step is due to equation 54. Since $(h_l)_k = \phi\left((h_{l-1})_k + (\boldsymbol{W}_l h_{l-1})_k\right)$, we can show $\boldsymbol{E}(h_l)_k^2 \ge (h_{l-1})_k^2$ given $h_{l-1}$ by numerical integral of Gaussian variable over an interval. Hence we have $\mathbb{E}\|h_l\|^2 \ge \boldsymbol{E}\|h_{l-1}\|^2 \ge \cdots \ge \boldsymbol{E}\|h_0\|^2 = 1$ by iteratively taking conditional

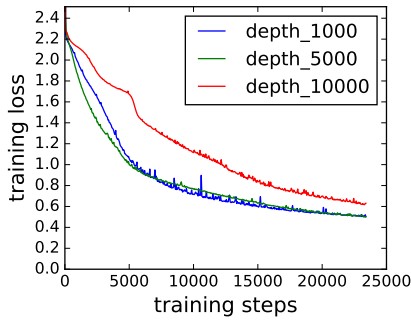 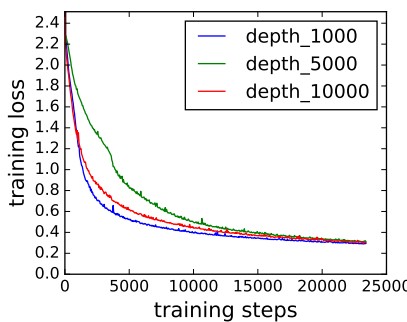

Figure 5: ResNet with 1000/5000/10000 layers. The width is 8 for the left figure and 16 for the right.

expectation. Then combined with equation 55 and the choice of $\tau = L^{-\frac{1}{2}+c}$, we have $\boldsymbol{E}\|h_{L-1}\|^2 \geq L^{2c}$. Because of the random Gaussian initialization of $\boldsymbol{W}_L$ and $h_L = \phi(\boldsymbol{W}_L h_{L-1})$, we have $\boldsymbol{E}\|h_L\|^2 = \|h_{L-1}\|^2$. Thus, the claim is proved.

$\square$

# H MORE EMPIRICAL STUDIES

| Dataset | Model | BLEU |
|---------|-------|------|
| IWSLT DE-EN | Transformer (Vaswani et al., 2017) | 34.8 |
| | Fixup (Zhang et al. (2019a)) | 35.0 |
| | Transformer + $\tau$ | **35.6** |
| WMT EN-DE | Transformer (Vaswani et al., 2017) | 28.4 |
| | Fixup (Zhang et al. (2019a)) | 28.1 |
| | Transformer + $\tau$ | **29.1** |

Table 3: Experiment results on machine translation task (Higher is better).

In this section, we train Transformer with $\tau$ but no normalization layer. We conduct experiments on two standard machine translation tasks: IWSLT DE-EN and WMT EN-DE. We modify the model by multiplying a fixed $\tau$ right after each residual addition and removing all the normalization layers. We do not use scalar bias in our Transformer+$\tau$ model. For IWSLT DE-EN, we adopt the Transformer-base model and set $\tau = 0.5$. For WMT EN-DE, we use Transformer-big model and set $\tau = 0.3$. Dropout is set as $0.5$ for base model. For big model, we set attention dropout and activation dropout as $0.1$. Other hyperparameters are the same as Vaswani et al. (2017). For Fixup, we use the implementation by Zhang et al. (2019a). All models are trained for $150$ epochs with $25k$ batch size. We average the checkpoints of the last 10 epochs and evaluate the BLEU score. The scores are shown in Table 3.

# I EXTREME DEEP RESNET

Per reviewer's request, we plot the training curves of extremely deep ResNets with the width $m \in \{8, 16\}$ and the depth $L \in \{1000, 5000, 10000\}$ in Figure 5. We set $\tau = \frac{1}{L^{0.5}}$. We see that even for ResNet with depth 10000 and width 8, the training curve although fluctuates, manages to converge. This reflects the depth dependency of ResNet is weak, echoing our stability result.

