# OpenReview forum: "STABILITY AND CONVERGENCE THEORY FOR LEARNING RESNET: A FULL CHARACTERIZATION"
_ICLR.cc/2020/Conference — Reject_

### Official Review · AnonReviewer1 · 2019-10-23
**Official Blind Review #1**

**Rating:** 3

**Review:**

This paper shows the stability of ResNet for the output scale of resblock: $\tau < \sqrt{L}^{-1}$ and shows the explosion for $\tau > L^{-0.5+c}$. Based on this analysis, a linear convergence rate of the gradient descent for the squared loss is also shown. In addition, this paper empirically verifies the efficiency of the initialization of $\tau = \sqrt{L}^{-1}$ with/without batch normalization.

Contributions of the paper are summarized below:
- Provides a sharp characterization of the largest scale of outputs of resblocks: $\tau$.
- Enlarges the class of ResNets with the global convergence guarantee of the gradient descent w.r.t. the scale of $\tau$. (A related study [Allen-Zhu+2018b] focused on the scale of $\tau = 1/L$).
- Improve the depth dependence in the network width and the iteration complexity compared to existing studies.

Significance:
To theoretically justify the success of ResNets, this line of research is important, and I think this paper makes a certain contribution in this sense, although there seems to be much room for improvements regarding the depth dependence in the network width. (Unrealistically high over-parameterization: $m=n^8L^7$ is still required). In addition, a proposed technique (Theorem 1) to derive gradient bounds is surprising and beyond the intuition. Indeed, a simple way of using a natural spectral bound cannot explain the stability of ResNets with $\tau=L^{-0.5}$.

Clarity:
The paper is well organized but there are some technical concerns as described below.

Quality:
I think the quality of the paper can be improved.
- Maybe probabilities in some statements are not correct, for instance, a probability in Theorem 2 should depend on the value of $c$.
- It is better to specify the value of $c$ in Theorem 1. It may take an arbitrary small value at the expense of the probability.
- In the convergence analysis of the gradient descent (Theorem 5), the dependency on $\tau$ should be mentioned. Is there no effect on the convergence rate by the choice of $\tau$?

Minor comments:
- The network architecture is different from the standard ResNets. Usually, non-linear activation functions are applied after resblocks. Can theoretical analyses be extended to such a setting?
- Initialization scales of parameters in related studies, as well as $\tau$, should also be clarified for a fair comparison of $\tau$.
- In proofs, citations of theorems and lemmas seem incorrect, e.g., Theorem 1 may be used in the proof of Lemma 3 (and Theorem 3 too?) instead of Lemma 1.
- Typo In Theorem 1: for all $a > l$  --> for all $b > a$ (?).
- There is no definition of $h'_{i,l}$ in Lemma 4. Is it a perturbation of a hidden node h?

-----
Update:
I thank the authors for the response and hard work.
I would like to keep my score. Some concerns have been resolved. Moreover,  I think highly of a proof technique (Theorem 1). However, the advantage of ResNets is still unclear because high over-parameterization is required.

**Experience Assessment:**

I have read many papers in this area.

**Review Assessment: Checking Correctness Of Derivations And Theory:**

I did not assess the derivations or theory.

**Review Assessment: Checking Correctness Of Experiments:**

I assessed the sensibility of the experiments.

**Review Assessment: Thoroughness In Paper Reading:**

I read the paper at least twice and used my best judgement in assessing the paper.

---

> ### Author Response · Authors · 2019-11-09
> **Response to Reviewer #1**
>
> We thank the reviewer for the feedback. The comments are addressed as follows.
>
> Quality:
>
> 1. The probability in Theorem 2 should depend on the value of c.
> A: The factor of the probability depending on “c” is clearly spelled out in the proof. We update the paper with explicit dependence on c in theorem statements.
>
> 2. It is better to specify the value of c in Theorem 1. It may take an arbitrary small value at the expense of the probability.
> A: We can set $c$ for a specific value for example $c=0.1$, which is sufficient for establishing the convergence theory in the downstream claims. Then Theorem 1 gives a condition on $m$ that $m >100(\log L)$.
>
> 3. In the convergence analysis of the gradient descent (Theorem 5), the dependency on $\tau$ should be mentioned. Is there no effect on the convergence rate by the choice of tau?
> A: We want to clarify this as follows. 1) The convergence analysis in Theorem 5 does not depend on $\tau$ explicitly since the gradient upper bound and lower bound are not tight enough to reflect the effect on the convergence rate by the choice of $\tau$. 2) Our result already improves the convergence of ResNet, since it “reflects the value of $\tau=1/\sqrt{L}$ determines sharply the trainability”.
>
> Minor comments:
>
> Q: The network architecture is different from the standard ResNets. Usually, non-linear activation functions are applied after resblocks. Can theoretical analyses be extended to such a setting?
> A: The non-linear activation location in the network architecture we studied is the same as that in the original ResNet paper (He et al. 2016). For the Preact ResNet, the residual block is the mapping $h_l = h_{l-1} +\tau W_l \sigma(h_{l-1})$. Our theoretical analysis carries over this setting without difficulty.
>
> Q: Initialization scales of parameters in related studies, as well as tau, should also be clarified for a fair comparison of tau.
> A: We take the usual He initialization $W_{i,j} \sim N(0, 2/m)$, the same as related studies (Allen-Zhu et al., Du et al.). Hence the comparison is fair. The initialization of Fixup method is set as suggested in the original paper.
>
> Q: In proofs, citations of theorems and lemmas seem incorrect, e.g., Theorem 1 may be used in the proof of Lemma 3 (and Theorem 3 too?) instead of Lemma 1.
> A: Thanks for pointing out this. We have corrected the citation error.
>
> Q: Typo In Theorem 1: for all $a > l$  for all $b>a$?.  No definition of $h’_{i,l}$ in Lemma 4.
> A: Thanks for pointing out these. We correct the typo and add the definition of $h’_{i,l}$. The paper has been updated with addressing the comments.

---

### Official Review · AnonReviewer2 · 2019-10-24
**Official Blind Review #2**

**Rating:** 1

**Review:**

The authors analyze the stability of a randomly initialized ResNet in terms of the scaling of the network and its depth. They claim to characterize the spectral stability of a randomly initialized network (and local perturbations of it), as well as that of the forward and backward process. These results, along with some further results, are used to extend convergence proofs of ResNets towards more standard initializations. Some of the claims are evaluated empirically.

Unfortunately, this paper is not a good contribution at the moment, as the work, although potentially interesting, is somewhat incremental and poorly presented. In particular, the results (including the proofs) only extend the results from Allen-Zhu et al, and are often hard to follow. The presented empirical results are nonetheless interesting and should be expanded upon. Here are some detailed comments

- The theorems should be stated more carefully, and in particular, care should be given to make precise which constant terms are hidden away in the big-O and big-Omega notations. E.g. in theorem 2 the probability must have some dependency on c hidden away, which should be made explicit, especially given the assertion “where c can be arbitrarily small”. Similarly, in theorem 1, it is claimed that c is constant, but the theorem is only non-vacuous for either m growing or c ~ log L. There are also some typos in theorem statements (e.g. missing O in theorem 3 and lemma 3), which can easily confuse the reader.

- Although I did not have time to check all the proofs, they should be treated with more care. I recommend avoiding big-O and big-Omega notations in the proofs as much as possible, as it can make some steps extremely confusing: in the proof of theorem 1, after “taking an \epsilon-net over …”, the probability does not change despite applying an union bound! The union bound argument should be more clearly spelled out, especially given the fact that some smoothness should also be established for this argument (e.g. the trivial sub-multiplicative bound).

- The empirical results are interesting, and I think could be taken even further. Indeed, the authors show promising results for learning a global scale parameter (in terms of performance), and it would be interesting to explore: 1) the value of the learned \tau after training: is it still of the order 1 / \sqrt{L}? Or does it take some other value. 2) whether it is better to have a \tau per layer or one for the whole network. The presentation of the current results could be improved by including standard deviation when averages are reported, and changing figure 4 with axes starting at zero (bar charts which do not start at zero are extremely misleading).

- There are also many typo and grammar issues. Although these do not seriously impede understanding, the paper could be improved by addressing those (at least running a spell-check).

**Experience Assessment:**

I have read many papers in this area.

**Review Assessment: Checking Correctness Of Derivations And Theory:**

I assessed the sensibility of the derivations and theory.

**Review Assessment: Checking Correctness Of Experiments:**

I assessed the sensibility of the experiments.

**Review Assessment: Thoroughness In Paper Reading:**

I read the paper at least twice and used my best judgement in assessing the paper.

---

> ### Author Response · Authors · 2019-11-09
> **Response to Reviewer #2**
>
> Thank the reviewer for the feedback. The concerns are clarified as follows.
> 1. Relation to Allen-Zhu et al. 2019 and Du et al. 2019.
> Indeed, there has been previous work studying the convergence of training ResNet, e.g., Allen-Zhu et al. 2019 and Du et al. 2019 and we study the same model as theirs. However, we give the full characterization for the convergence of training ResNet in terms of the values of $\tau$, while previous work only builds a looser convergence for $\tau<1/L$. Our result of stability for the case $\tau<1/\sqrt{L}$ is new and nontrivial, while in previous work the stability under condition $\tau<1/L$ is obvious as a naïve spectral norm bound applies. Furthermore, our theory characterization, i.e., value of $\tau=1/\sqrt{L}$ has nice practical guidance.
> In short, our contribution is significant as it provides full characterization for the convergence of learning ResNet in theory and gives practical guidance on ResNet design by immediately connecting the theory to practice, while Allen-Zhu et al. does not.
>
> 2. (1) The theorems should be stated more carefully. Some constant terms are hidden away in the Big-O and Big-Omega notations, e.g., in Theorem 2, the probability must have some dependency on “c” hidden away. (2) In Theorem 1, the theorem is only non-vacuous for either m growing or c ~ log L. (3) Typos in theorem statements, e.g., missing O in theorem 3 and Lemma 3.
> A: (1) We are indeed stating our theorems with full alertness. In Theorem 2, How the probability depending on “c” is clearly spelled out in the proof.  To get the claimed probability and bound, it is required that $c>\ tau\sqrt{L}\sqrt{\log23}$. We note that the constant c is irrelevant to the values of $L$ because $\tau\sqrt{L}=O(1)$. We can choose the coefficient of $\tau$ small enough to enable $c$ small.  We make the dependency of $c$ explicit in the theorem statements in the updated paper. (2) This is not at all an issue. Theorem 1 states the high probability bound of the spectral norm and how the probability scales with $m, c$ and $L$, which produces a condition on $m$ that $m >(1/c^2) * (\log L)$. As usually done in non-asymptotic analysis, we first state the conditions on $m$ and finally take the intersection over all collections of conditions to guarantee the theorems simultaneously. (3) There is no missing $O$ in Theorem 3 and Lemma 3.
>
> 3. Avoid using big-O and big-Omega notations in the proofs as much as possible. In the proof of theorem 1, after “taking an \epsilon-net over …”, the probability does not change despite applying a union bound!
> A: We add in the updated paper a full procedure of $\epsilon$-net argument. A new factor ($m\log 23$) induced by $\epsilon$-net is hidden in $\exp(-\Omega(mc^2))$. We try our best to make the argument clear enough with necessary exposure of detailed math formula while maintaining the readability.
>
> 4. The empirical results are interesting with room to improve (1) the value of the learned \tau after training: is it still of the order 1 / \sqrt{L}? Or does it take some other value. (2) whether it is better to have a \tau per layer or one for the whole network. (3) include the standard deviation and change the figure 4 with axes starting at zero.
> A: (1) Thanks for pointing out this question. The initial values of $\tau$ for ResNet 20/32/56/110/1202 are $\{0.33, 0.26, 0.19, 0.13, 0.04\}$ and the learned values of $\tau$ are $\{0.38, 0.35, 0.33, 0.31, 0.27\}$. However, if setting the initial values to be the learned values, the deep ones explode. (2) We have tried to have a separate $\tau$ each layer but it does not give better performance than a shared $\tau$. (3) We update the paper with the reviewer’s suggestion.
>
> 5. The paper is updated with careful  grammar checking.

---

### Official Review · AnonReviewer3 · 2019-10-25
**Official Blind Review #3**

**Rating:** 6

**Review:**

The paper presents a non-asymptotic analysis of ResNet stability which determines a 'cutoff' value for the residual block scale factor characterizing stability vs output explosion, and improves upon prior results by finding a larger range for the scale factor that lead to global convergence in non-convex optimization. Theoretical findings are corroborated via experiments confirming the validity of the 'cutoff' value. Additional experiments are conducted indicating that using the cutoff value, ResNet can be trained even without normalization layers, and that the cutoff value is also beneficial with normalization as it allows effective training of very deep ResNet.

This is an interesting submission which presents important theoretical results while also showing their practical pertinence via experimental validation.

The paper is well written and the presentation is clear. Prior work is reviewed appropriately.  The reviewer found it particularly informative to provide intuition and present a proof sketch next to each theoretical results.

The convergence analysis leads to a depth dependence of ResNet and the authors claim that this is not a *real* dependence and  attributes it to bounding techniques handling non smooth activation. The authors further indicate in their experiments that for a given width the convergence of ResNet does not depends *much* on depth, compared to feedforward networks. So there might still be a dependence though much milder than that found by the current theory.  It would be interesting to more precisely characterize the dependence observed in experiments.

The authors claim that their analysis leading to the non-asymptotic bound on the spectral norm of the feedforward process via martingale theory might be relevant for other problems. It would be useful if the authors could elaborate and in particular indicate if their technique could carry to analyse other structures beyond ResNet.

Minor comments:
"What else values" ---> Are there other values
"our first claim is a new spectral norm" -->  is a new bound of the spectral norm
"we does no make such" --> we do not make




**Experience Assessment:**

I have read many papers in this area.

**Review Assessment: Checking Correctness Of Derivations And Theory:**

I assessed the sensibility of the derivations and theory.

**Review Assessment: Checking Correctness Of Experiments:**

I carefully checked the experiments.

**Review Assessment: Thoroughness In Paper Reading:**

I read the paper thoroughly.

---

> ### Author Response · Authors · 2019-11-09
> **Response to Reviewer #3**
>
> Thanks a lot for your review. We address the concerns as follows.
> 1. As for the depth dependence, the stability result (Theorem 1) shows that the forward and backward processes are stable if the width $m$ scales with $\log L$ while a similar result for vanilla feedforward network (Allen-Zhu et al.2018) requires the width $m$ scales at least with $L\log L$.
> The convergence result (Theorem 4) states that the depth dependence is polynomial with $L$, which comes only from the semi-smoothness characterization (Theorem 6). As we discussed in the paragraph after Theorem 6, the depth dependence term in the semi-smoothness characterization is relatively small to other terms under the over-parameterized regime e.g. $m>L$ (we note that this does not mean that $m$ scales with $L$ linearly). Hence the depth dependence in the convergence result is much weaker than the case of feedforward network.
> For experiments, we tried much deeper ResNet in Appendix I and the training curves is similar to that in Fig. 1 as a $\log $ term dependence is hard to reveal in experiments.
>
> 2. It was pointed out by others that our result and proof may be used to show the spectral norm of a product of i.i.d. matrices that is called the top Lyapunov exponent, which might have interesting applications in stochastic stability in addition to the convergence analysis of gradient descent for ResNet. We believe this is suitable for future study.
>
> 3. We thank the reviewer for pointing out the typos and grammars. The paper is updated with careful grammar checking. We wish the reviewer could reconsider the evaluation of our paper if the comments are addressed well.

---

### Public Comment · ~Devansh_Arpit2 · 2019-10-06
**Inaccurate description of our paper**

Thank you for citing our work “How to Initialize your Network? Robust Initialization for WeightNorm & ResNets”. Given your work provides an in-depth analysis of the same initialization for ResNets that we derived in our paper (tau=1/sqrt(L) ), I think it is important to mention that the current description of our work in your paper is not fully accurate. You mention that in our derivation of initialization scheme for ResNets, we do not provide any theoretical justification for ||f(h)|| = ||h||, where f(.) is a residual block. This is not true. This condition is true because the weights of a residual block are initialized such that the residual block is norm preserving. We have derived this initialization for weight normalized networks in our paper. Further, our initialization scheme for ResNets works irrespective of the presence or absence of normalization layers as long as the two conditions you reported are met. And this condition is indeed true for un-normalized network with ReLU activation under He initialization (see “The Benefits of Over-parameterization at Initialization in Deep ReLU Networks”).

These comments aside, I found the theoretical results of your paper (especially on convergence using 1/sqrt(L) initialization) interesting (disclaimer: I have not looked through the proofs). The experimental results are also promising. I think it is great that an initialization much simpler than Fixup works better in practice.

---

> ### Author Response · Authors · 2019-10-07
> **Accurately,  the paper [1] assumes that ||f(h)|| = ||h|| with justification for E ||f(h)||^2 = ||h||^2.**
>
> Thanks for the interest in our paper.
> Our current description is accurate and may miss detailed discussion. For norm preserving, [1] has argued   E ||f(h)||^2 = ||h||^2 for weight normalized networks and [2,3] have argued (1-\epsilon) ||h||<||f(h)||<(1+\epsilon) ||h|| for  He initialization, where f(.) is a feedforward layer mapping. However, this by no means implies the assumption||f(h)|| = ||h|| in Theorem 3 of [1]. We will give more detailed discussion about the contribution in [1] in next revision.
>
>  In contrast, we does not make such stringent assumption and provide rigorous non-asymptotic analysis (high probability bound) for the stability for the forward/backward pass with τ = 1/\sqrt(L)  for the standard initialization scheme used in practice.
>
> [1] Devansh Arpit, Victor Campos, and Yoshua Bengio. How to initialize your network? Robust initialization for weightnorm & resnets. In Advances in Neural Information Processing Systems (NeurIPS), 2019.
> [2] Zeyuan Allen-Zhu,Yuanzhi Li,and Zhao Song. A convergence theory for deep learning via over-parameterization. arXiv preprint arXiv:1811.03962, 2018b.
> [3] Devansh Arpit and Yoshua Bengio. The Benefits of Over-parameterization at Initialization in Deep ReLU Networks. Arxiv 2019.

---

> > ### Public Comment · ~Devansh_Arpit2 · 2019-10-08
> > **I think there is a misunderstanding**
> >
> > In [1], we study initialization in an asymptotic setting (infinitely wide network). Thus our analysis of expectation of norm holds with arbitrary precision as network width tends to infinity. Similarly, the bounds derived in [3] also hold with arbitrary precision as width tends to infinity. So our assumption is justified.

---

### Author Response · Authors · 2019-11-10
**paper has been updated**

 Dear reviewers,
We have uploaded the paper after taking the suggestions. The main changes are as follows.
1. Rephrase Theorem 1 and Theorem 2, make the dependency on constant $c$ clearer.
2. We add a full $\epsilon$-Net argument in the proof of Theorem 1 (Appendix B).
3. Correct the typos and missing notations.

---

### Decision · Program_Chairs · 2019-12-19

**Decision:**

Reject

**Comment:**

The article studies the stability of ResNets in relation to initialisation and depth. The reviewers found that this is an interesting article with important theoretical and experimental results. However, they also pointed out that the results, while good, are based on adaptations of previous work and hence might not be particularly impactful. The reviewers found that the revision made important improvements, but not quite meeting the bar for acceptance, pointing out that the presentation and details in the proofs could still be improved.